

# Spatial and temporal distributions of surface mass balance between Concordia and Vostok stations, Antarctica from combined radar and ice core data : First results and detailed error analysis.

Le Meur Emmanuel[1], Magand Olivier[1], Arnaud Laurent[1], Fily Michel[1],
Frezzotti Massimo[2], Cavitte Marie[3], Mulvaney Robert[4], and Urbini Stefano[5]

[1]University Grenoble Alpes, CNRS, IGE, Grenoble, France
[2]ENEA, Agenzia Nazionale per le nuove tecnologie, l'energia e lo sviluppo sostenibile, Rome, Italy
[3]Institute for Geophysics, University of Texas at Austin, Texas, USA
[4]British Antarctic Survey, Cambridge, UK
[5]Istituto Nazionale di Geofisica e Vulcanologia, Rome, Italy

*Correspondence to:* Le Meur Emmanuel (emmanuel.lemeur@univ-grenoble-alpes.fr)

**Abstract.** Results from Ground Penetrating Radar (GPR) measurements and shallow ice cores carried out during a scientific traverse between Dome Concordia (DC) and Vostok stations are presented in order to infer both spatial and temporal characteristics of snow accumulation over the East Antarctic plateau. Spatially continuous accumulation rates along the traverse are computed from the

identification of three equally spaced radar reflections spanning about the last 600 yr. Accurate dating of these Internal Reflection Horizons (IRHs) is obtained from a depth-age relationship derived from volcanic horizons and bomb testing fallout on a DC ice core and shows a very good consistency when tested against extra ice cores drilled along the radar profile. Accumulation rates are then inferred by accounting for density profiles down to each IRH. For this latter purpose, a careful error

analysis showed that using a single and more accurate density profile along a DC core provided more reliable results than trying to include the potential spatial variability in density from extra (but less accurate) ice cores distributed along the profile.

    The most striking feature is an accumulation pattern that remains constant through time with persistent gradients such as a marked decrease from 26 mm w.e.yr$^{-1}$ at DC to 20 mm w.e.yr$^{-1}$

at the South West end of the profile over the last 234-yr average (with a similar decrease from 25 mm w.e.yr$^{-1}$ to 19 mm w.e.yr$^{-1}$ over the last 592 yr). Insights into the time-dependency reveal a steady increase of accumulation on the East Antarctic plateau during the last 600 yr and more particularly since about 200 yr as already suggested by previous studies relying on GPR and/or time markers in ice cores. Maximum margins of error are in the range 4 mm w.e.yr$^{-1}$ (last 234 yr) to

2 mm w.e.yr$^{-1}$ (last 592 yr), a decrease with depth mainly resulting from the time-averaging when





computing accumulation rates. The gradients proposed in this study remain however significant since only the reduced non systematic component (with regard to space or time) of these error terms has to be considered when interpreting spatial or temporal trends.

# 1  Introduction

The surface mass balance (SMB) over the Antarctic ice sheet is of primary interest for many purposes. The mass budget of the Antarctic ice sheet can be seen as the difference between all mass exchange with the atmosphere over all the ice sheet surface (i.e the SMB) minus the calving of icebergs and basal melting under ice shelves at the coast. The first term is globally positive as it represents snow accumulation over most parts of the ice sheet with the exception of blue ice and wind

crust areas with negative SMB due to wind scouring and/or sublimation (Bintanja, 1999; Scambos et al., 2012). The second is negative despite the possibility of marine ice accretion under some parts of the shelves. Although current changes in the mass budget of Antarctic ice are believed to result mainly from recent dynamical effects leading to an enhanced flow of ice through outlet glaciers and ice streams (e.g., Rignot and Kanagaratnam, 2006), changes in surface mass balance should not be

disregarded since they potentially operate over large areas. Numerical predictions by Krinner et al. (2006) suggest that advection of warmer saturated air masses (hence containing more moisture) could lead to a 32 mm water equivalent per year increase for the Antarctic future SMB in the next century corresponding to a negative sea level contribution of $1.2 \mathrm{~mm.yr}^{-1}$ by the end of the 21st century. The SMB of the grounded Antarctic Ice sheet (AIS) is approximately 2100 $Gt.yr^{-1}$, with

a large interannual variability. Those changes can be as large as 300 $Gt.yr^{-1}$ and represent approximately 6% of the 1989-2009 average (Van den Broeke et al., 2011). Moreover, the SMB uncertainty is estimated to be more than 10% (equivalent to nearly 0.6 $mm.yr^{-1}$ of sea level rise ), which is at least equal to the ice discharge uncertainty (Frezzotti et al., 2007; Magand et al., 2007).

Measuring the SMB over ice sheets therefore represents a challenge which addresses a wide num-

ber of scientific glaciological and environmental issues. It first allows for a direct assessment of the SMB pattern and specific features with regard to time as well as spatial variability. Spatial variability of SMB operates over various scales ranging from less than a meter to several hundreds of km (see for instance a detailed description in Eisen et al. (2008)). Using SMB measurements for inferring mass budgets whether over a single drainage basin or all over the continent therefore requires some sam-

pling and interpolation strategies. Unfortunately, these approaches remain approximate and suffer from the sparsity of data especially over the Antarctic plateau. Remote sensing can be an alternative, but measured quantities are not always a reliable proxy for SMB, making ground truthing still necessary. Modeling is also a possibility but similarly requires control from field ground measurements and the more of these control points, the more accurate the results obtained. Persisting discrepan-



cies between spatially interpolated measured data (Arthern et al., 2006) and modelling results (e.g., van den Broeke et al., 2006) also tend to call for denser field measurements.

Accessing the temporal variability of SMB is also of major interest. As pointed by Eisen et al. (2008) knowing past and present conditions of SMB is necessary for predicting its behaviour under future climatic conditions. Moreover, independent measurements of past SMB are of importance for interpreting ice cores. Last but not least, SMB is one of the main factors driving ice dynamics and as such, represents the principal boundary condition for ice sheet models. If past SMB is obviously required for modeling past ice sheet dynamics, it appears also necessary for assessing the current dynamical state of the ice sheet given the fact that the corresponding characteristic time response makes the present-day ice sheets still react to SMB changes that occurred for the past centuries or even millennia.

The proposed study aims to document both spatial and temporal variability of the SMB over a so-far unexplored region of the Antarctic plateau. Corresponding field work took place during a scientific traverse between the French-Italian Concordia station ($75,10°S$, $123.33°E$) and the Russian Vostok station ($78,49°S$, $106.65°E$) during the 2011/2012 austral summer (See Fig. 1). This traverse is one component of IPY lead project TASTE-IDEA (Trans-Antarctic Scientific Traverses Expeditions Ice Divide of East Antarctica ) and is also linked to the international SCAR project ITASE (Mayewski et al., 2005). The measurement technique is based on the complementary approaches of snow radar (GPR) and ice core analysis. The former provides a continuous mapping of isochronous layers leading to a spatialization of snow accumulation over long distances. On the other hand, ice cores yield a much more local information, but nevertheless necessary as a means of providing both time markers for dating radar horizons and density vertical profiles from which snow quantities accumulated over the different IRHs can be assessed. The complementarity also comes from the fact that local accumulation as revealed by ice cores has a limited representativeness due to the small scale variability of some processes leading to the net accumulation (wind driven sublimation, wind scouring, see for instance (Frezzotti et al., 2007)).

In the present study, data acquisition is first described with the radar data summarized under a merged profile whose post processing allowed the identification of three equally spaced IRHs covering the last $592$ yr. Ice core data are then presented with a focus on the depth markers obtained from chemistry analysis (volcanoes) and/or radio-isotope counting (nuclear tests) leading to the $4$ age-depth relationships later used for dating purposes. Special attention was paid to the vertical density distribution contained in the different ice cores of the project and used for computing the cumulative mass deposited above each of the three IRHs. Then follows a detailed error budget on the final computed accumulation rates which accounts for the three main sources of error stemming from the uncertainties in (i) density profiles, (ii) age and (iii) depth of the IRHs. Finally, spatial and temporal variations of the accumulation rates along the measured profile are presented, discussed and compared (when possible) with results from previous studies.

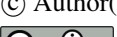


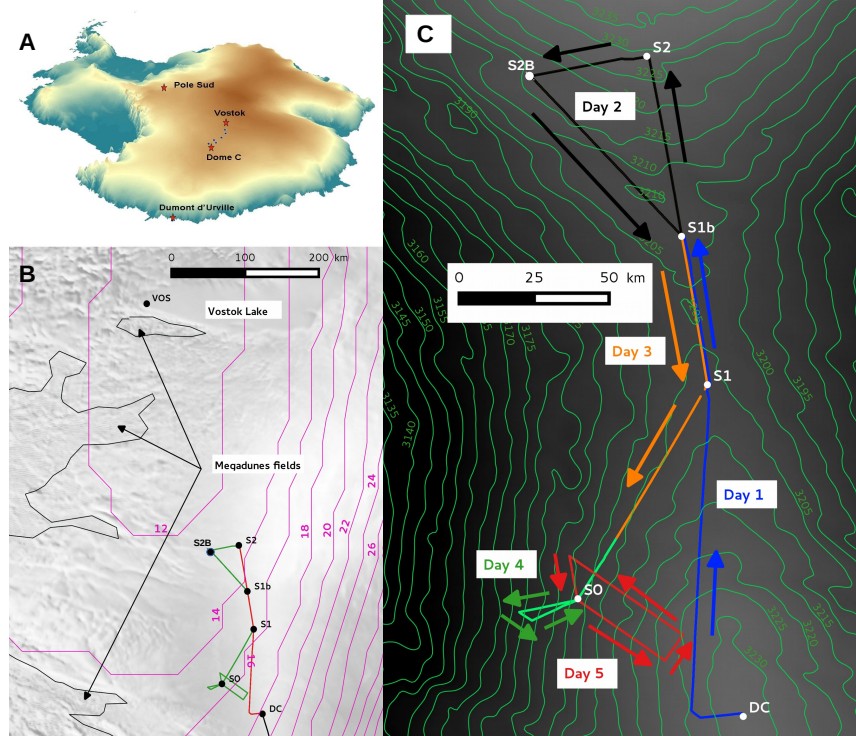

**Figure 1.** Map of the route followed by the 2011/12 TASTE-IDEA traverse over the east Antarctic plateau between the Dome Concordia (DC) and Vostok stations. (A) locations of departure and destination base stations on the plateau. (B) the portion of the traverse along which radar profiles have been acquired. (C) zoom detailing the five days of GPR profiling as well as the drilling sites of DC plus the locations of the firn cores retrieved during the traverse (S1, S2, S2B, S0) and used for dating the isochrones and deriving density profiles (see Sections 3 and 4). The pink contours in (B) represent iso-values of the precipitation rate as computed by the ERA 40 model of the European Center for Medium-Range Weather Forecasts (Simmons et al., 2007). Although the observed spatial and temporal variability is correctly reproduced, the precipitation is systematically underestimated (Genthon et al., 2015) and results require a systematic multiplying factor of about 1.5 to become realistic (Libois et al., 2014). Processes like sublimation and snow redistribution are omitted in the physics of the model and also explain the discrepancy with accumulation rate data.

## 2 Ground Penetrating Radar (GPR) data

### 2.1 Technical background and functional principle of GPR

The principle of GPR consists of measuring the two-way travel times of parts of an electromagnetic
95 pulse emitted toward the ground that are reflected on discontinuities within the observed medium.





When pulses are emitted at points along a profile, lateral continuity of the physico-chemical transitions within the medium yields wiggles of similar travel time (hence depth) along the resulting radar trace. Once put side by side, these wiggles express the form of horizons also called IRHs (Internal Reflection Horizons) visible on the recorded radargram. In this respect, GPR is analogous to seismic reflection but with an electromagnetic wave instead of an acoustic wave.

Discontinuities responsible for these partial reflections result in changes in the complex dielectric constant $\epsilon*$ of the medium. For the snow pack, essentially two main factors can be responsible for these changes as explained in (Eisen et al., 2008) and references therein. The first concerns the real part, namely the permittivity, which varies with density whereas the second acts on the imaginary part which is sensitive to the conductivity of the medium. As a result, radar horizons within the first few hundreds meter of firn and ice are essentially due to changes in density and acidity from volcanic deposits.

The huge advantage of the IRHs comes from their isochronous nature, although this has not been rigorously (physically) demonstrated, only heuristically by connecting ice core chemistry analysis of different drill sites (e.g., Eisen et al., 2004; Frezzotti et al., 2005)) or by comparing accumulation rates obtained form both IRH depths and stake lines (Vaughan et al., 2004). This isochronity is however understandable as it results from processes operating at the surface at the same time, such as acidic atmospheric deposits or density changes from the metamorphism of snow (at least the one initiated by surface interaction with the atmosphere like radiation crusts, wind redistribution). However, the physical processes leading to these IRHs still remain poorly known. For instance, as pointed by Eisen et al. (2008), annual layers are still visible with radar wavelengths much larger than the annual thickness (a typical $100-\mathrm{MHz}$ wave in the cold dry snow of the plateau has a wavelength of $2\ \mathrm{m}$ much larger than the $10\ \mathrm{cm}$ annual snow thickness at DC for instance) which contradicts the theory unless constructive interferences are considered as suggested by Palli et al. (2002) for instance.

Assuming isochronity, the spatial variations of snow accumulation can be obtained from the varying depth of the IRHs. It thus implies transforming the two-way travel times of the reflected waves into actual depths. This is only possible if the wave velocity within the medium is known all the way from the surface down to the specific IRH and the time radargram is properly migrated (see Section 2.3). Inferring accumulation rates then requires the association of an age to the IRHs from independent means such as correct recognition of fallouts from identified volcanic and/or bomb testing events along firn/ice cores for instance (see Section 4).

## 2.2 GPR set up

For the present project, GPR data were acquired with a MALÅ® ProEx GPR equipment fitted with a 100 MHz 'Rough Terrain Antenna (RTA)' towed behind one of the tractors used during the traverse with a constant separation of $2.2\ \mathrm{m}$ between the emitter and receiver parts. The apparatus, operating



frequency and set up are very similar to those used on a similar traverse between base stations of Dumont d'Urville and DC during the 2008/2009 austral summer (Verfaillie et al., 2012). The triggering was set to $1$ s meaning a radar trace every $4$ m or so given an average speed of $14$ km.h$^{-1}$

for the convoy. The time window was set to $1\mu s$ (allowing for an investigation depth of some $100$ m) and was sampled at $1.1$ MHz leading to roughly $1100$ samples per trace. During acquisition a first improvement of the signal-to-noise ratio was obtained by an up to 64-fold stacking (which, given the time window of $1\mu s$, remained compatible with acquisitions made every second). Moreover, the system was connected to a GPS receiver mounted on the vehicle and recording the geographic

position of every single trace along the profiles.

### 2.3    Resulting GPR radargrams

Profiles outlined in Fig 1 represent five days of measurements between 13/01/2012 and 17/01/2012 and led to an almost continuous record of $630$ km. The corresponding post-processed and concatenated radargram is shown in Fig 2. The post processing sequence consisted of time zero corrections,

a zero-phase low-cut filter (devow) to remove direct continuous currents, and an 'Energy decay' gain to compensate for the volumetric spreading signal attenuation. Band pass filtering reduced to a low-pass filter under the form of a spatial averaging over 50 traces which, given the huge number of traces (164 000 altogether), appeared more efficient than traditional Finite Impulse Response (FIR) filters. Time to depth conversion was obtained by migrating the radargrams with the help of a verti-

cal velocity profile for the radar wave. By investigating a similar region with similar firn properties, we followed the approach of Verfaillie et al. (2012) which consisted of only considering the effect of the firn density on the electromagnetic wave velocity as a result of a dry and clean snow over the Antarctic plateau. From the empirical relation of (Kovacs et al., 1995) relating permittivity and density, the wave velocity $c$ is obtained according to :

$$c = \frac{c_v}{1. + 0.845 \times \rho} \tag{1}$$

with $c_v$ the wave velocity in vacuum and $\rho$ the firn density (relative to water density). The density profile comes from a recent core drilled at DC (Leduc-Leballeur et al., 2015) where high resolution measurements allowed for a detailed profile (see Section 5.2). The true restitution of dipping reflectors theoretically requires a topographic migration, but this latter appears only necessary where the

topographic variations are of the same order as the reflector depths. Given the extreme flatness of the investigated area of less than $2$ m.km$^{-1}$ (topography of the upper surface and hence that of the IRHs) no such correction has been applied here. Last, ice sheet dynamics potentially changing the IRHs geometry has to be accounted for, except in the upper part of the ice/firn column, especially over areas of slow motion such as over the Antarctic plateau as in the present case (Eisen et al.,

165    2008).



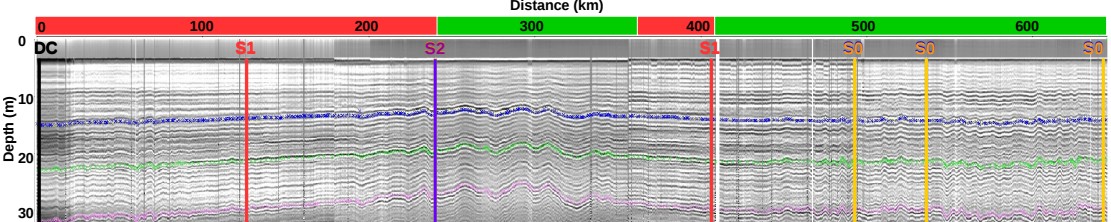

**Figure 2.** Continuous post-processed radargram corresponding to the merged profiles as outlined in Fig. 1. The numerous observable IRHs result from as many reflections within the snow pack and are each characterized by a laterally varying depth and a specific age. The coloured (blue, green and pink) aligned dots emphasize the three most evident and continuous IRHs that are used in the present study. The radargram as depicted here (restricted to the first 35 m of firn) has undergone a time migration (see Section 2.3) meaning that the vertical axis represents true depths instead of two-way travel times. The apparently shifted vertical scale results from the time-zero correction. The horizontal axis is the cumulated distance along the profile as given by the GPS data. The white vertical stripe is due to a breakdown of the GPR leading to a gap of less than 2 km in the data. Also represented are the crossing points with the ice/firn cores DC, S1, S2 and S0 where the intersecting depths with the three selected IRHs have been inferred for checking their isochronicity (see Section 4). The red and green upper banner refers to the position with respect to the topographic divide and correspond to the color convention of the bottom right of Fig. 1 (red for 'divide' and green for 'off divide' locations).

## 3 Shallow cores

### 3.1 Ice core drilling

Four cores were used to time calibrate the IRHs observed with ground-penetrating radar; one intermediate ice core (VOLSOL 1, 102-m deep) collected at DC during the 2010-11 Antarctic summer

season (VOLSOL program (Gautier et al., 2016)) and three shallow ice cores (S0, S1 and S2, around 20-m deep each) retrieved along the TASTE-IDEA traverse. The ice cores were collected using two different electromechanical drilling systems (10-cm diameter for VOLSOL program and 5.8-cm diameter for TASTE-IDEA). The recovered core pieces were sealed in polyethylene bags in the field and stored in clean isothermal boxes before treatment. The VOLSOL-1 ice core was treated

at DC using the laboratory facilities for chemical studies whereas the TASTE-IDEA shallow firn cores were transported in frozen state to the cold-room facilities of the Laboratory of Glaciology (IGE) in Grenoble, France, for radiochemical and chemical studies. At DC, the ice core was cut in cleaned conditions and samples were sealed in precleaned tubes before ion chromatography analysis. In Grenoble, sample preparation was also performed under clean-room laboratory conditions.

After stratigraphic observations and measurements of bulk density, the four firn cores were divided into two half cores. One half was dedicated to radioactivity measurements, and the other half was



analysed by ion chromatography. Due to the lack of seasonal variations of any chemical or physi-
cal parameters in snow between DC and Vostok stations, year-by-year dating of the snow layers is
impossible. Only specific reference horizons can be used. In our study, the chronology of IRHs was
established with the aid of sulfate spikes concentrations from past volcanic events coupled with the
identification of recent thermonuclear bomb testing marker levels of CE 1955 and 1965.

### 3.2   Sample preparation and analysis

Samples for chemical and radiochemical analysis were prepared with a time resolution of about
$0.3 - 0.4$ yr (2-cm length) and $1.5 - 2.0$ yr (10 cm), respectively, which is necessary for the detection
of volcanic eruptions and radioactive levels. The samples were prepared and analysed using stringent
contamination control procedures.

For the VOLSOL-1 ice core treatment at DC, only sulfate concentrations were determined. They
were measured (more than 5000 samples) on a Metrohm 850 Professional IC system coupled with
a Seal XY-2 Sampler. A Thermo Scientific Dionex IonPac AS11-HS (4-mm diameter, 50-mm long)
anion separation column was used. Such a short column allowed us to develop a fast (2 mn per
sample) isocratic method using a 7 mM NaOH solution as eluent. Measured concentrations were
calibrated in the $10 - 1000$ ng.g$^{-1}$ range using dilutions of a commercial 1000 µg.g$^{-1}$ sulfate stan-
dard solution.

For the TASTE-IDEA firn cores treated at IGE, one half core was treated to measure major and
organic chemical species from the surface down to 20 m deep (more than 5000 samples for all cores).
Chemical analysis were performed using two different systems, enabling cross-checking of possible
contamination processes and also validate the sulfate concentrations results. Indeed, the first mea-
surements series were done using the same device system as the one used at DC for the VOLSOL-1
ice core treatment (sulfate concentrations measurements only) and the second series (same samples)
was analysed with a Dionex© ICS3000 dual ion chromatography system. This last chromatography
system was setup for the analysis of cations ($Li^+$, $Na^+$, $NH_4^+$, $K^+$, $Mg^{2+}$, $Mn^{2+}$, $Ca^{2+}$, $Sr^{2+}$)
and anions ($F^-$, $Cl^-$, $MSA^-$, $NO_3^-$, and $SO_4^{2-}$) down to sub-ppb level and high level accuracy
(6 standards calibration, relative standard deviation $< 2\%$). The two different chemical analysis sys-
tems found no contamination and were reliable for sulfate concentration determinations. The results
of test series consequently allowed use of only the first system with short sample run analysis (1.5
mn versus 22 mn) to provide complete and rapid sulfate profiles as needed.

The second half of the TASTE-IDEA firn cores were processed for artificial radioactivity mea-
surements ($^{90}Sr$, $^{137}Cs$, and to a lesser degree $^{241}Am$, daughter of $^{241}Pu$) by a Berthold© B770
low-level beta counting and Canberra© very-low background BEGe gamma spectrometry device.
Analysis were performed with a continuous sampling every 10 cm from the surface down to 6 m,
i.e. the necessary depths to detect radioactive reference horizons from 1950s to 1980s atmospheric





thermonuclear bomb tests. Details of the sampling and measurements procedures are given in Mag-and (2009), Loaiza et al. (2011) and Verfaillie et al. (2012).

### 3.3 Identification of volcanic eruption signals and volcanic chronology

Although sulfate in Antarctic snow comes from sea-salt spray and to a lesser degree from crustal erosion (Maupetit and Delmas, 1992; Cole-Dai et al., 2000), and principally from atmospheric oxidation of biogenic dimethylsulphide (DMS) emitted by oceanic phytoplanktonic activity (Saltzman, 1995; Prospero et al., 1991), volcanic eruptions are also major sources of $SO_4^{2-}$ during active eruptions. To identify volcanic signals in ice cores, it is then necessary to caclculate the non-sea-salt sulfate

(nss$SO_4^{2-}$) corresponding to total sulfate minus sea-salt sulfate, and set a threshold above which spikes can be attributed to volcanic deposition. Over the East Antarctic plateau areas, part of the sulfate background that could be related to continuous emissions from non-explosive volcanic activity, seems to be minor (Patris et al., 2000; Cole-Dai et al., 2000; Castellano et al., 2004). The sea-salt sulfate contribution to total sulfate budget, evaluated using $Na^+$ as a specific marker (Palmer et al.,

2001; Röthlisberger, 2002), is less than $10\%$. The Holocene crustal contribution, calculated by non-sea-salt $Ca^{2+}$ as a continental dust marker (Röthlisberger, 2002; Castellano et al., 2004; Plummer et al., 2012), is even lower ($< 0.05\%$). Since these contributions are of the same order of measurement reproducibility, we did not correct sulfate concentrations and will not distinguish between total sulfate and non-sea-salt sulfate in the following discussion. However, particular attention was paid to

develop a reliable method for distinguishing volcanic signals from non-volcanic sulfate background and an outline of the method used is described in Cole-Dai et al. (1997, 2000); Stenni (2002); Castellano et al. (2004) and Igarashi et al. (2011). The procedure employed to identify candidate signals of volcanic eruptions in the sulfate profiles of the different cores follows that of Gautier et al. (2016).

Between five and eight volcanic events were identified in each TASTE-IDEA firn cores and ten

events were detected in the VOLSOL-1 ice core from the surface to 20 and 30 m, respectively, through the application of the above described procedures (see summary in Table-1). In the considered depths for shallow TASTE-IDEA cores and the VOLSOL-1 ice core, the main volcanic events in the $1601 - 2012$ CE period are Pinatubo (Philippines, 1992 CE), Agung (Indonesia, 1964 CE), Krakatau (Indonesia, 1885 CE), Coseguina (Nicaragua, 1836 CE), Tambora (Indonesia, 1816 AD),

Unknown (1809 AD), Jorullo-Taal (Philippines, 1758 CE), Serua (Indonesia, 1695 CE), Gomkonora (Indonesia, 1676, CE) and Huyanaputina (Peru, 1601 AD). The sulfate deposition of Kuwae eruption (Vanuatu, $1457 - 1458$ AD (Salzer and Hughes, 2007; Sigl et al., 2013, 2014, 2015)) could be actually dated as 1459 AD, i.e., 5 to 6 years later than previously assumed (Gao et al., 2006, 2008).

The temporal duration of volcanic signals in the ice has been evaluated by several studies (Cole-

Dai et al., 2000; Palmer et al., 2001; Castellano et al., 2004) and general temporal durations range from 1 to 3 years. In this work, the temporal duration of volcanic signals lagged between 1.2 (3 consecutive samples) and 6 years (15 consecutive samples).





**Table 1.** Depths of the dating events along the mentioned cores from known volcanoes as well as thermo-nuclear bomb testing and used for producing the time-depth relationships as depicted on Fig. 3

| Event name | Date of emission | Date of deposition | DC [a] VOLSOL-1 | S0 | Explore [b] | S1 | S2 |
|---|---|---|---|---|---|---|---|
| Pinatubo | 1991 | 1992 ±1 | 1.51 (0.08) | - [c] | - | - | 1.45 |
| '1965 Bomb Tests' | 1962-64 | 1965 ±1 | 4.20 [d] | 3.67 | - | 3.37 | 2.75 |
| Agung | 1963 | 1964 ±1 | 3.81 (0.08) | - | - | - | - |
| '1955 Bomb Tests' | 1952-54 | 1955 ±1 | 4.60 [d] | 4.26 | - | 4.00 | 3.36 |
| Krakatau | 1883 | 1885 ±2 | 8.81 (0.07) | - | - | - | - |
| Coseguina | 1835 | 1836 ±1 | 11.99 (0.07) | 11.34 | 10.48 | 11.31 | 10.04 |
| Tambora | 1815 | 1816 ±1 | 12.91 (0.07) | 12.44 | - | 12.37 | 10.98 |
| Unknown | ? | 1809 ±3 | 13.33 (0.07) | 12.72 | - | 12.83 | 11.30 |
| Jorullo-Taal | 1754 | 1758 ±1 | 15.99 (0.06) | 15.02 | - | - | 13.72 |
| Serua | 1694 | 1695 ±1 | 19.29 (0.06) | 18.14 | - | 18.15 | 16.48 |
| Gamkonora | 1673 | 1676 ±3 | - | 19.12 | - | 19.39 | 17.26 |
| Huaynaputina | 1600 | 1601 ±2 | - | - | - | - | 20.20 |
| Unknown | ? | 1646 ±3 | 21.92 (0.06) | - | 21.65 | - | - |
| Kuwae | 1457 | 1459 ±3 | 30.19 (0.05) | - | 27.35/27.61 [e] | - | - |
| Unknown | ? | 1259 ±3 | - | - | 34.94 | - | - |

[a] The value in parenthesis represents the depth shift to apply as the core was extracted a year earlier (see Sec. 4.2); [b] Time markers of the S0 and Explore cores have been merged to produce the hybrid depth relationship of FIG. 3 (see also text in Sec. 4.2); [c] An hyphen indicates an undetected volcano or an unmeasured nuclear fallout ; [d] These two bomb test fallouts were actually measured on a nearby shallow core close to the VOLSOL-1 one in 2011/12 and served in the determination of its age/depth relationship ; [e] These two peaks respectively dated CE 1453 and 1460 have actually been measured in the Explore core and attributed to the Kuwae volcano (Mulvaney, personal communication).

### 3.4 Artificial radionuclides deposition over Antarctica and associated chronology

Artificial radioisotopes resulting from atmospheric thermonuclear tests carried out between 1953
and 1980 were deposited in Antarctica after transport in the upper atmosphere and stratosphere,
creating distinct radioactive reference levels in the snow. The dates of arrival and deposition in this
polar region are well known and therefore provide a means to estimate Antarctic snow accumulation
rates or describe air mass circulation patterns (Pourchet et al. (2003); Magand (2009) and references
therein). The 1955 and 1965 CE radioactivity peaks provide two very convenient horizons for dating
snow and firn layers and thus measuring accumulation. Special techniques have been developed
over the last 40 years to detect and measure artificial and natural radionuclides present in the ice
sheets (Pourchet et al. (2003); Magand (2009) and references therein). In Antarctica, $^{90}Sr$, $^{241}Pu$
(deduced from $^{241}Am$ analysis) and $^{137}Cs$ radionuclides constitute the well-known debris products
of atmospheric thermonuclear tests between CE 1953 and 1980 that we could still use to identify
the two well-known reference layers in snow (1955 AD and 1965 AD peaks), corresponding to





the arrival and deposition of artificial radionuclides in this region. Total beta counting and gamma spectrometry remain the most frequent radioactivity measurement device systems used to clearly detect artificial radionuclides and unambiguously determine the 1955 CE and 1965 CE peaks depths in the cores. The $1955 \pm 1$ C E and $1965 \pm 1$ CE peaks were both identified in each TASTE-IDEA

firn core and in an extra shallow core close to the VOLSOL-1 one (see Table-1).

## 4  IRH dating

### 4.1  Identification of major IRHs

Interpretation of radar data first consisted of identifying contrasting IRHs whose continuity could be tracked all along the entire merged profile. Three equally spaced IRHs were selected within the first

35 m of firn (no sufficiently clear reflectors could be used any deeper) and are emphasized with associated colors in Fig. 2. In order to properly capture the spatial variability, picking of these reflectors was made every km or so yielding each time a depth and the corresponding geographical position (coloured dots on the figure). Of interest is the phasing of these three IRHs which is characteristic of a stationary accumulation spatial pattern. It comes from the fact that local extrema in accumulation

keep the same positions through time, allowing for cumulative effects with time which amplify IRH undulations with depth (Verfaillie et al., 2012).

### 4.2  Methodology for dating the IRHs

Dating is achieved by detecting the depths at which the reflectors intercept (or pass very close to) an ice/firn core where a depth/age relationship has been obtained (see Section 3). In the present case,

as can be seen from Figs 1 and 2, each of the three reflectors passes over several ice core locations and sometimes several times over the same one, thereby offering redundant dating possibilities. As a consequence, one core was selected as the reference and served for proposing an age *ab initio* for each of the three IRHs. By providing time/depth markers down to about 30 m, allowing for a good constraint of the quadratic fit down to the deepest reflector ($r^2 > 0.99$), the DC VOLSOL-1

ice core (Gautier et al., 2016) was chosen as the reference and the remaining ones (S0,S1 and S2) then allowed for a check of the isochronous character of the IRHs (see Fig. 3). The choice for the DC core is also justified by the numerous available ice core dating schemes for the site which gives a good level of confidence in the proposed age/depth relationships (Parrenin et al., 2007).

For all of these cores, volcanic markers obtained from sulfate spikes have been complemented by

the identification of the 1955 CE and 1965 CE radioactive reference levels (see two points around 4 m deep on Fig. 3). However, the limited depths of the extra S0, S1 and S2 cores (around 20 m) make their corresponding quadratic fits questionable for depths below the second IRH. This limitation was partly remedied thanks to the companion program Explore which benefited from the logistics of the traverse and provided a deeper (110 m) ice core extracted some 70 m away from




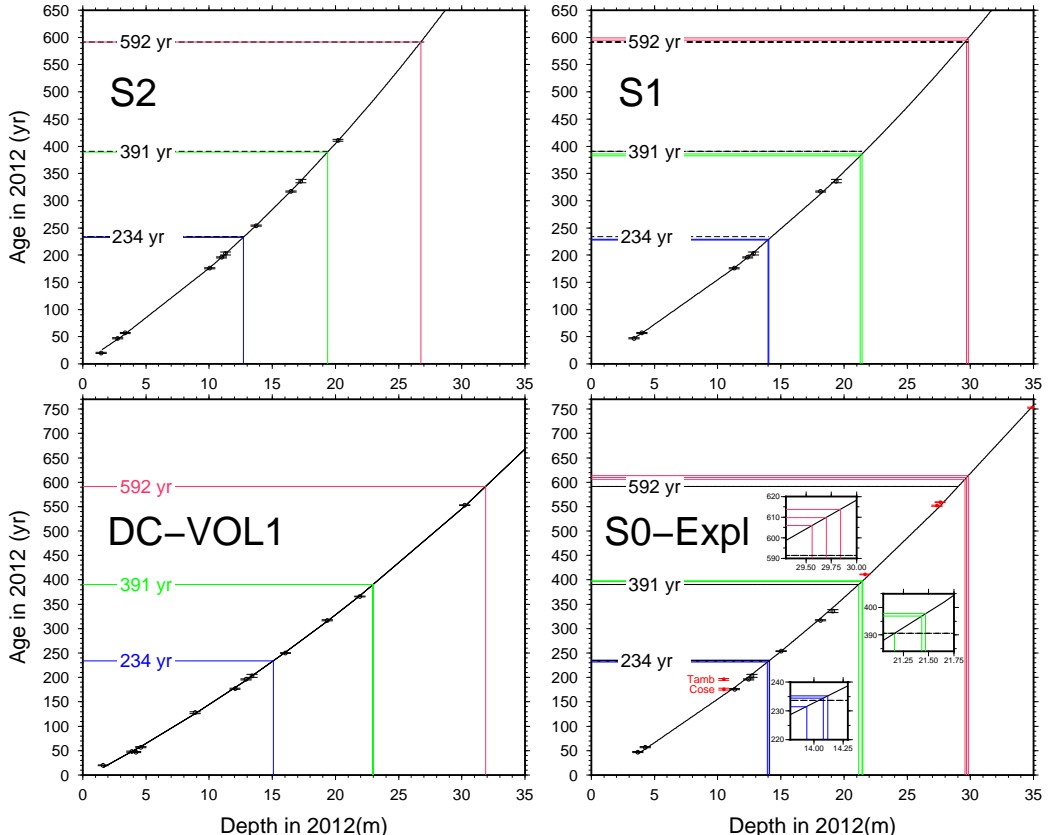

**Figure 3.** Identified and dated deposits placed in an age/depth representation according to their depths within the DC-VOLSOL-1 (DC-VOL1), S0-Explore (S0-Expl), S1 and S2 ice cores along the GPR profile. Almost systematically similar dated events (black symbols) were used from one core to the other (with vertical error bars barely exceeding 2 yr) except for the longer DC-VOLSOL1 core which allowed for one extra marker down to 30 m (Kuwae volcano) and the S0-Explore Core (Huaynaputina, Kuwae (2 peaks) and unknown-1259 CE) for which extra time markers are in red. A degree-2 polynomial fit (black line, $r^2 > 0.99$) has then been used as a continuous dating curve permitting to date any intermediary depth. At the DC-VOLSOL-1 reference core, depths of $15.08$ m, $22.97$ m and $31.89$ m respectively give ages of $233.74$ yr, $390.54$ yr and $591.71$ yr for the three IRHs (later rounded to $234$ yr, $391$ yr and $592$ yr). For the other cores, intercepting depths of the three IRHs are reported and corresponding ages proposed that can then be compared to the reported reference ages form DC (black dashed lines). Insets on the S0 curves are enlargements allowing for an assessment of the dispersion in the IRH depths and resulting ages when crossing the core three times (note that S1 is only crossed two times).





the S0 core. Sulfate peaks originating from volcano deposits were analysed by continuous melting
and Fast Ion Chromatography and provided extra time markers down to about 35 m (in red on the
figure, Mulvaney, personal communication, see also Table 1). As a consequence, the S0 core has
been supplemented with these deep markers to build a hybrid S0-Explore age-depth relationship as
represented in Fig 3.

Having been extracted during the VOLSOL program (2010/11 field season) the DC-VOLSOL-1
core is a year older than the S1, S2 and S0-Explore cores. As a consequence, a depth correction
has been applied to each IRH volcanic depth in the form of a downward shift of 8 cm (modern
snow-equivalent accumulation for DC) weighted by the density ratio between the relevant depth and
the surface (i.e. 8 cm for the surface reducing to 4.53 cm for the Kuwae volcano at the depth of
31.20 m). Moreover, integration of the two extra 1955 and 1965 thermonuclear horizons along this
core was possible from dedicated measurements along an extra core drilled at DC in 2012 during the
scientific traverse.

### 4.3   Age results

Corresponding measurement points and resulting quadratic depth/age calibrations for the four cores
are depicted in Fig. 3. As can be seen, ages for each IRH remain consistent along the entire radar
profile with a limited scatter at the various crossings (either between cores or when passing several
times at the same core site). For instance, the uppermost blue IRH varies by only 3% among all
crossing points and has a maximum discrepancy of only 6 yr (2.5%) with the DC reference age
(see detailed figures in Table 2). As for the second IRH (green), the scatter is 16 yr (4%) with
a maximum deviation of 9 yr from the DC age of 391 yr. Lastly, the deepest one (pink) has a
maximum deviation of 3.75% (22 yr) with the reference age which also corresponds to its total
dispersion. The only noticeable discrepancy is to be found with the deepest IRH when passing the
S0 site where crossing depths lead to systematically older ages. One explanation could lie in the
loss of continuity for this deepest IRH towards the end of the profile where it sometimes becomes
very faint (see bottom right of Fig. 2). The other possibility could result from the hybrid depth-age
relationship obtained at S0 from two different cores. Because of the short-scale variability of surface
snow post-deposition processes (Eisen et al., 2008), the distance of 70 m between the two cores
may explain such discrepancies as was observed by Gautier et al. (2016) who found depth shifts
of the order of 20cm between five cores only separated by 1 m at the DC site. They also come to
the conclusion that logging errors and analysing procedures may play a role, especially if different
coring and analysing teams are involved as was the case for the S0 and Explore cores. Although it
can be criticised, the construction of such an hybrid age-depth relationship makes some sense in our
case since we only considered the extra Explore markers at depths where no other volcanoes were
available in the initial S0 core (> 20 m). The resulting quadratic fit still yields $r^2$ greater than 0.99.





**Table 2.** Depth, corresponding age and deviation with respect to the DC -VOLSOL-1 reference age for the three
IRHs whenever they intersect the DC-VOL-1, S0-Expl, S1 and S2 cores

|  | DC | S0-1 | S0-2 | S0-3 | S1-1 | S1-2 | S2 |
|---|---|---|---|---|---|---|---|
| IRH1 depth | 15.08 | 14.08 | 13.94 | 14.12 | 13.97 | 14.05 | 12.70 |
| IRH1 age | 234 | 234 | 232 | 235 | 228 | 229 | 233 |
| IRH1 error | - | 0% | 1% | 0.5% | 2.5% | 2.25% | 0.5% |
| IRH2 depth | 22.97 | 21.47 | 21.43 | 21.16 | 21.43 | 21.27 | 19.38 |
| IRH2 age | 391 | 398 | 397 | 391 | 386 | 382 | 390 |
| IRH2 error | - | 1.75% | 1.5% | 0% | 1.25% | 2.25% | 0.25% |
| IRH3 depth | 31.88 | 29.56 | 29.70 | 29.84 | 29.84 | 29.70 | 26.76 |
| IRH3 age | 592 | 606 | 610 | 614 | 599 | 595 | 592 |
| IRH3 error | - | 2.25% | 3% | 3.75% | 1.25% | 0.5% | 0% |

335 A sulfate peak observed in the Explore core a the depth of $10.48$ m (red symbol labeled 'Tamb'
on the figure) was initially attributed to Tambora (1816 AD) which markedly conflicted with the S0
Tambora depth of $12.44$ m. After confirmation that such a major Volcano as Tambora could easily
be missed in a low accumulation area as pointed out in (Gautier et al., 2016)) where this latter is
missing in two out of the five VOLSOL cores, it seemed more reasonable to attribute this peak to
340 the Coseguina volcano (1835 AD, see red symbol 'Cose') for which the depth of $11.34$ m (black
symbol) has been found in the S0 core leading to the proposed quadratic fit (black curve in Fig. 3).
Considering a second quadratic fit (not represented on the figure) in which Coseguina is placed at
$10.48$ m led to age increases of only 2 and 1 year for respectively the upper and middle IRH and had
no significant impact on the age of the lowest.

345 **4.4 IRH age uncertainty**

The two main factors that can potentially alter the isochronous character of the picked IRHs are (i)
errors in the building of the age-depth relationship at each coring site and (ii) errors in the depth
estimates of the IRHs at these same sites deduced from the radargrams. As for the first, owing to
the short-scale changes in snow deposition and erosion as expressed by the surface roughness, the
350 archiving process can sometimes be significantly different for neighbouring cores thereby raising the
problem of the correct representativeness of a single core. For several volcanoes Gautier et al. (2016)
found differences in the depths of the corresponding sulfate peaks by as much as $\pm 20 cm$ from one
core to the other among a set of five different cores all separated by a meter. Major volcanoes such
as Tambora are even missing in two of the five cores. As a consequence, on top of errors in the
355 correct depth measurements of the chemical peaks, surface processes lead to an uncertainty in the
core-derived age-depth relationships. Figure 4 reports the time markers along the five cores from the
study of Gautier et al. (2016). From quadratic fits along each core, five different ages are computed



for our IRH depths of $15.08$ m, $22.97$ m and $31.89$ m yielding respective RMS deviations of $6$ yr, $7$ yr and $7$ yr with respect to our reference ages of $234$ yr, $391$ yr and $596$ yr. These deviations can be considered as good estimates of the uncertainty in the correct representativeness of the single cores of our study. The residency time of volcanic aerosols in the stratosphere (from 2 to 4 yr) is variable from one volcanic event to the other and therefore also contributes to some uncertainty in the depth-age relationship which makes an overall uncertainty of $8$yr on the age for a given depth, a result very similar to that of Eisen et al. (2004).

As for the depth accuracy of the picked IRHs, it mainly results from the physics of radar and the chosen frequency for the electromagnetic wave. Eisen et al. (2004) gives a review of all possible resultant sources of errors and their results are applicable to our case because of the use of the same commercial radar set from MALÅ® Geoscience Sweeden, with the only difference that they use 200 and 250 MHz frequencies instead of our 100 MHz. One first source of uncertainty lies in the thickness the 100 MHz-wave can resolve. Although it theoretically amounts to $\lambda/4$, one fourth of the wavelength in the firn, $\lambda/2$ is usually considered as more realistic which in our present case gives $1$ m. Additionally, the pulse width also matters (length of the energetic part of the source wavelet) and was determined by Common Mid Point (CMP) analysis to be $12$ ns with our system (Verfaillie et al., 2012) leading to a tracking accuracy of half this duration, which in the firn gives a shift of some $1.2$ m. Since these two errors do not systematically add up, one generally considers the worse of the two. Finally, interpreting radargrams and picking reflectors is partly subjective and operator dependent which leads us to estimate the overall uncertainty on the IRH depth to be $1.5$ m at the worse. Given a relatively constant slope of $20$ yr.m$^{-1}$ of the age-depth relationships (see figs 3 and 4), this depth uncertainty maps into an age uncertainty of about $30$ yr. This uncertainty in the depth positioning nevertheless has a strong systematic component (in other words the error in depth positioning remains mainly the same along a given IRH) and therefore only partly alters the isochronous character of the reflectors. The same can be said for depth errors resulting from a wrong assessment of the vertical velocity profile. Only lateral variations not properly accounted for would contribute, but it is well known that the plateau firn only undergoes minor lateral variations in its depth distribution of density (Muller et al., 2010) and hence velocity at the scale of a couple of hundred km.

As a result, one can reasonably argue that a true isochronous IRH can theoretically deviate by 10 to 15 yr. When comparing to the RMS dispersions of respectively $3$ yr, $6$ yr and $13$ yr for IRHs 1 to 3 at the crossing points as revealed by Figure 3, we come to the conclusion that our three IRHs are actually isochronous.

We nevertheless have to keep in mind that the absolute error on the age (to be used in the budget error for mass balance computation, see Section 5.7) remains potentially higher than these 10 to 15 yr by integrating the biases from systematic incorrect positioning and errors from time to depth conversion in the radargrams.

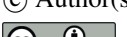


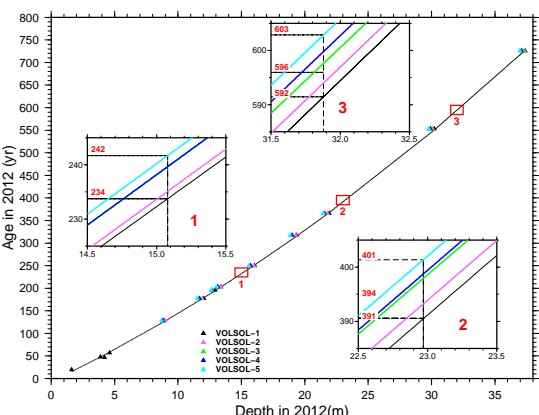

**Figure 4.** Time markers along the five cores as studied in (Gautier et al., 2016). The reference core is VOLSOL-1 (see Section4.2) for which the quadratic fit is entirely represented in black, whereas insets represent enlargements where all quadratic fits allow for an assessment of the age dispersion around the reference at the depths of our three IRHs.

## 5 Snow accumulation along the GPR profile

### 5.1 Accumulation computation

Assuming an IRH at a given depth $H$, the total mass $M$ of the unit surface (1 m by 1 m) firn column above it can be expressed as :

$$M = \int\limits_{0}^{H} \rho(z)\,\mathrm{d}z \tag{2}$$

where $M$ can be considered as a mass flux per $m^2$ because of the implicit unit surface. It therefore expresses as $kg.m^{-2}$ which, given the density of water, also represents millimeter water equivalent ( mm w.e.). The average accumulation rate over the period since the deposition of the IRH is finally obtained by dividing by the age of the IRH leading to a final result in $mm\,\text{w.e.yr}^{-1}$.

### 5.2 Firn density profiles for accumulation rate computations

Similarly to the depth and age, proper estimation of the density profile down to a given IRH is crucial for a good assessment of the corresponding accumulation rate. If spatial variations of accumulation mainly result from differences in snow cumulative heights as revealed by undulating reflectors, one can reasonably wonder to which extent geographical changes in density may also impact. Unfortunately, over the Antarctic plateau, density profiles are usually concentrated at limited sites with almost no reliable data in between. In the present case, density profiles from deep ice cores are only available at the two 'extremities' of the radar line (DC and S0). Some extra shallow cores were also

(c) Author(s) 2017. CC BY 3.0 License.





**Table 3.** Density cores at DC along with the characteristics of their quadratic fits. Cumulated masses (kg.m$^{-2}$) down to each of the three IRHs at respective depths of 15.08, 22.97 and 31.88 $m$ are computed from the raw data as well as from the fitted curves.

| Core name | Depth (m) | $r^2$ | RMSE (kg.m$^{-3}$) | cum. raw 1 15.08 $m^{(a)}$ | cum. fit 1 15.08 $m^{(a)}$ | cum. raw 2 22.97 $m^{(a)}$ | cum. fit 2 22.97 $m^{(a)}$ | cum. raw 3 31.88 $m^{(a)}$ | cum. fit 3 31.88 $m^{(a)}$ |
|---|---|---|---|---|---|---|---|---|---|
| DC5 [b] | 81.01 | 0.988 | 13.51 | 6060 | 6054 | 10087 | 10058 | 15129 | 15101 |
| DC3 | 81.32 | 0.987 | 13.88 | 6050 | 6033 | 10103 | 10045 | 15154 | 15101 |
| Firetracc | 50.06 | 0.939 | 24.76 | 6352 | 6321 | 10350 | 10343 | 15361 | 15362 |
| Itase-98 | 44.17 | 0.973 | 17.39 | 5963 | 5894 | 9937 | 9832 | 14884 | 14788 |
| Itase-DC1 | 17.47 | 0.905 | 19.95 | 6176 | 6171 | - [c] | - | - | - |
| Itase-DC2 | 17.90 | 0.976 | 9.48 | 6218 | 6217 | - | - | - | - |
| Itase-DC3 | 19.27 | 0.901 | 20.52 | 6119 | 6062 | - | - | - | - |
| Itase-DC4 | 18.15 | 0.948 | 13.68 | 6250 | 6264 | - | - | - | - |

[a] in kg.m$^{-2}$ ; [b] (Leduc-Leballeur et al., 2015) ; [c] depth not reached by the core

drilled along the radar profile (S1, S2 , S2B) but their maximum depths (about $20$ m) are limiting at least for interpreting the two deepest IRHs. However, owing to the fact that over the plateau meteorological parameters controlling the densification process (e.g. air temperature, wind activity, solar

radiation..) exhibit small gradients at the regional scale, a relative uniformity in firn density profiles can be first assumed over distances of the order of $100$ km as can be anticipated from the small divergence between the DC and S0 profiles (see Section 5.4). The limited variability was confirmed by Fujita et al. (2011) from pits and surface snow measurements during the JASE traverse and more specifically the resulting small impact on accumulation rates was demonstrated by Ruth et al. (2007)

by comparing water equivalent depth profiles between Dome Fuji and EPICA DML. This led us in a first instance to only consider deep ice cores from the DC site along which precise and high resolution density profiles have been performed (e.g., Leduc-Leballeur et al., 2015). A later sensitivity test is proposed (see Section 5.5) consisting of assessing changes in accumulation expected from integrating the contribution of density profiles from different locations. These changes are then

compared to the uncertainties solely arising from the choice of a given density depth profile (correct representativeness, measurement errors, etc..) to be exclusively used in the accumulation rate computation all along the radar line (see Section 5.6).

**5.3 Density profiles at DC**

In the framework of the EPICA project (Augustin et al., 2004), along with the main core, several

extra shallower drilling projects have yielded as many density profiles over at least the first $50$ m for the DC area (see Table 3). A first question as to which of these cores to use for computing accumulation rates arises, or in other words, what can be the sensitivity of the results to the choice




between different cores at a given site. Figure 5 shows the depth density profiles for each of the cores listed in the table along with the corresponding cumulative mass with depth. The cumulative masses

have been computed either from the raw data (not represented on the figure) or from the quadratic fits (see corresponding results at the three IRH depths in the table). Despite a pronounced scatter in the individual data points around their respective fits (see Fig. 5 and associated RMSE in the table), considering raw or fitted data does not significantly change the depth-cumulated mass with differences not exceeding $1.15\%$ at the most (ITASE-98 at the depth of IRH1).

This illustrates a symetric scatter of data points above and below the fit which leads to an almost exact compensation when summing up the cumulative mass. As a consequence, fitted density curves will hereafter be considered in the computations of accumulation rates. The second result comes from the limited dispersion in the cumulative mass from one core to the other as can be easily seen from the figure. The maximum discrepancy is to be found between the Itase-98 and Firetracc ice

cores with relative differences of $7\%$, $5\%$ and $4\%$ at the depths of respective IRHs 1, 2 and 3.

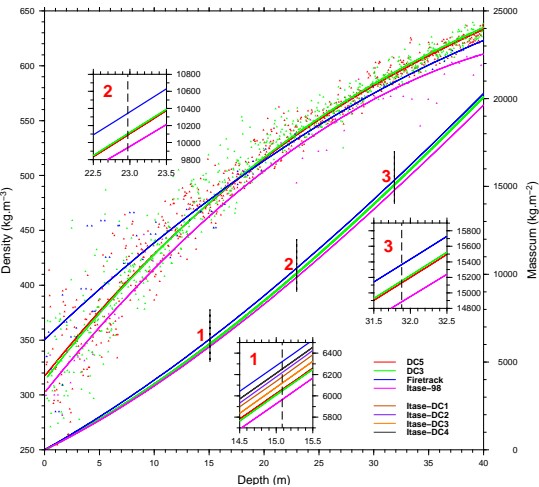

**Figure 5.** Density profiles (left scale) as mentioned in the text and detailed in table 3. Also shown is the corresponding cummulative mass with depth (right scale). For the sake of clarity, only cores (and associated cumulative masses) exceeding 20 m are fully represented (DC5, DC3, Firetracc and Itase-98 ice cores). Insets correspond to enlargements around the depths of IRHs 1 2 and 3 where all mass curves are depicted and allow for an estimation of the dispersion in cumulative masses.

Of interest is the almost perfect overlap of the DC3 and DC5 curves which come from the same project and have been measured according to the same protocol by the same operator. This tends to show that :

(a) Most of the difference between different cores results from measurement errors





(b) The variability of the density expressing at the scale of the typical distance between these cores
(from $m$ to $km$ scale) is of stochastic type (Libois et al., 2014) and is therefore rapidly canceled
out by the depth averaging process of cumulative mass computation leading to accumulation
rates. Indeed the DC3 and DC5 cores are separated by $1.5$ km but still show remarkable consis-
tency in their quadratic fits or in their associated cumulative mass distribution with depth.

(c) Provided that systematic error measurements can be minimized, a single density profile should
then be considered as representative of the local accumulation rate.

The very good similarity of the DC3 and DC5 cores attests to the quality of density measurements
as confirmed by the novelty and strictness in the measurement protocols (Leduc-Leballeur et al.,
2015). This led us to choose the DC5 as the reference for our accumulation rate computations. A
sensitivity to the various density profiles along the radar line is later proposed in Sec. 5.5.

### 5.4 Density profiles along the traverse route

Fig. 6 now shows the density profiles from some of the cores situated along the traverse route. Apart
from the DC5 core (Leduc-Leballeur et al., 2015) all other cores were drilled in the framework of
the project (with the S0-Expl core resulting from the Explore companion program) and correspond-
ing density profiles measured in the following years. What the figure shows is the good similarity
between the density values along the DC and S0 cores apart from slightly lower values for the latter
over the first 30 m. This result is not surprising given the limited distance between the two sites
(66 km), although DC is located on a dome, whereas S0 is already on the eastern side of the DC-
Vostok ridge (see Fig. 1) where the slope in the prevailing wind direction can have a significantly
different impact on the wind-driven erosion/redeposition of snow (Frezzotti et al., 2002) and hence
on the density. Going further south (S1,S2 and S2B cores) does not either change the density values
significantly (at least over the first 20 m) although the comparison is less straightforward given the
marked dispersion in the first meters.

The observed variability in density along a core as observable in Fig. 6 comes from the natural
variability due to depositional processes and from measurement errors. These latter mainly result
from measurement difficulties due to the weak cohesion of snow in the first meters preventing from
obtaining well defined shapes that are later weighted as part of the density calculation process. As a
consequence, an uncertainty of $\pm 20$ kg.m$^{-3}$ over the first 7 m linearly decreasing to $\pm 15$ kg.m$^{-3}$
at 15 m deep and remaining constant further down is proposed for these density measurements
(see error bars on the figure). These uncertainties are less pronounced than the ones proposed by
Verfaillie et al. (2012) ($10\%$ of the density value) at least for the deepest parts of the cores, since it
corresponds to more recent cores over which extra-careful and high resolution density measurements
were performed.



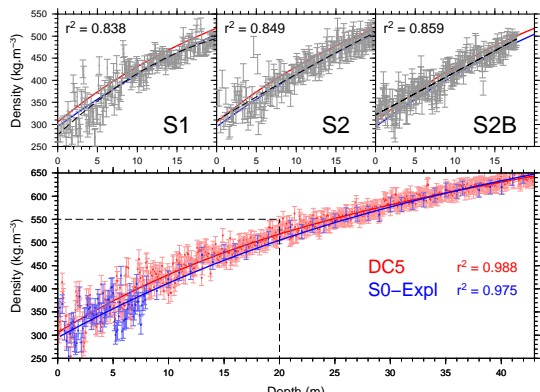

**Figure 6.** Density measurements along the ice/firn cores from Dome Concordia (DC5) and from the traverse route (S0-Expl, S1, S2 and S2B). For DC5 and S0-Expl, a degree-3 polynomial offered the best fit and is represented by a continuous line. The DC core used here (DC5) was drilled in 2012 and was chosen for the high resolution in density measurements (more than 1400 samples over the first 80 m). The S0-Expl core was drilled in the frame of a companion program (program Explore which benefited form the logistics of the traverse, and provided density measurements down to 110 m (Chappelaz, personal communication). For the sake of clarity, and having a much reduced depth of about 20 m, the S1, S2 and S2B cores are depicted in the upper insets whose imprint corresponds to the bottom dashed square. Their quadratic fits (black dashed line) as well as those of DC and S0 (respectively red and blue lines) are also shown.

As for the natural dispersion, it is partly explained by the seasonality of deposition and changes in the local climatic conditions (Hörhold et al., 2011) and more significantly by wind-driven erosion/redeposition processes (Frezzotti et al. (2002); Libois et al. (2014)). Indeed, snow redistribution by wind is the result of a subtle interaction between the wind field and the small scale surface topography eventually leading to short-scale variations in the physical properties of the surface snow. Libois et al. (2014) showed that a stochastic scheme of the snow erosion/deposition by the wind in a multilayer detailed snowpack model could explain a significant part of the near surface variability in snow density. However Hörhold et al. (2011) show that due to competing mechanical processes, this variability rapidly decreases with depth to reach a minimum at around $600 - 650\,\mathrm{kg.m^{-3}}$ before increasing again towards a second local maximum.

Despite compensation effects due to the random character of the natural variability with depth as observed in the cumulated masses (Section 5.3) measurement errors, on the other hand do not systematically cancel with depth integration (especially if systematic) and can thus become a significant contributor to the overall uncertainty due to density. Although difficult, an attempt to assess this term is proposed in the budget error Section. Before doing so, it appeared interesting to first assess the changes in accumulation pattern to be expected from the choice of different density profiles and then




to compare them to the accumulation error resulting from the total uncertainty on a given density
profile. Such a comparison should provide insights into the relevant strategy to adopt.

### 5.5 Sensitivity of the accumulation rate to different density profiles

Figure 7 shows the three accumulation rates derived along each of the three IRHs along the radar
line from the density profile provided by the DC-5 reference core. Since the computation uses a

single density profile, the IRHs and the resulting accumulation curves show very similar undulations
and particularly, the persisting overall shape from one IRH to the other is maintained expressing an
almost perfectly stationary pattern through time. Of importance is the pronounced gradient when
following the ridge going from DC towards S2 (see map on Fig. 1) with a more than $20\%$ loss within
250km.

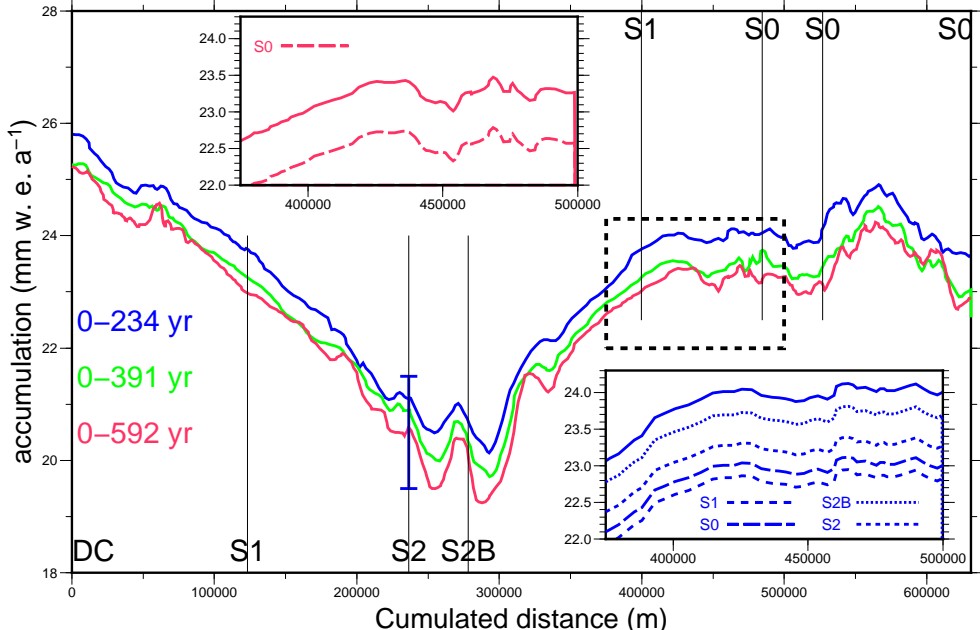

**Figure 7.** Computed accumulation rates since the deposition of the three IRHs according to the polynomial fit
of the DC-5 density profile. The lower inset represents an enlargement over the dotted square and shows the
sensitivity of the accumulation rate at the uppermost IRH to the use of the DC-5, S0, S1, S2 and S2B cores
whereas the upper one does the same but for the lowermost IRH and with only the DC-5 and S0 cores. Ranges
for the accumulation rate at S2 inferred from the SR50 sensor (see Sect. 6.1) are represented by the blue error
bar along the S2 line.

As for the sensitivity to the the use of density profiles from various locations, it expresses under
the form of a uniform shift along a given IRH (as can be seen from the insets) but with a decreasing





sensitivity with the depth of the IRH. The difference in accumulation rates from the use of the DC and S0 densities amounts to $1\,\mathrm{mm\,w.e.yr^{-1}}$ for the 234-yr IRH (lower inset) and reduces to about half of it when considering the 592-yr reflector (upper inset). The explanation is to be found in the

rapidly converging fits of the DC and S0 density curves with depth (Fig. 6). Indeed, the increasing cumulative difference in mass is more than counterbalanced when dividing by the age of the IRH in the accumulation rate computation process. Using the less reliable S1, S2 and S2B does not lead to a significantly higher sensitivity. With density fits in between those of the DC and S0 cores (see upper insets of Fig 6) S2 and S2B accumulation rates normally lay in between those of DC and S0. Only

the S1 accumulation curve exhibits a slightly larger difference due to a density curve systematically below those of DC and S0. It should be noted that because of their limiting depths of some 20m the sensitivity to the S1, S2 and S2B curves cannot be assessed down to the two lowest IRHs. The main conclusion to retain is a maximum deviation of the order of $1\,\mathrm{mm\,w.e.yr^{-1}}$ to expect from accounting for the geographical changes in density along the profile.

### 5.6 Proper choice of the density core to use

Assessing the uncertainty on the accumulation rates solely arising from measurement errors in the density profile used is not straightforward since part of these errors are random and certainly compensate with depth, a least partially. Moreover, small-scale accumulation variability that influences the representativeness of a single core for a given location also potentially contributes, even if lim-

ited in amplitude as was shown in Section 5.3. Because they result from different projects implying different measurements protocols, the numerous cores drilled in the DC area provide a means of estimating the order of magnitude of the combined effect of these two terms. In particular, the dispersion between all these cores should implicitly account for the most of the measurement errors and would only miss a systematic component (i.e an identical error for all measurements) which in any case

must remain small. Computation of the cumulative mass down to IRH1 for these 8 cores listed in Table 3 gives an average of $6127\,\mathrm{kg.m^{-3}}$ with a standard deviation of $141\,\mathrm{kg.m^{-3}}$. When dividing by the age of the IRH (234 yr) one obtains $26.18\pm0.60\,\mathrm{kg.m^{-3}yr^{-1}}$ or $26.18\pm0.60\,\mathrm{mm\,w.e.yr^{-1}}$. Similar computations down to the two deepest IRHs (but only concerning the first four cores) respectively give $25.75\pm0.54\,\mathrm{mm\,w.e.yr^{-1}}$ and $25.49\pm0.40\,\mathrm{mm\,w.e.yr^{-1}}$ for IRH 2 and 3. This

decrease in the uncertainty results from a stabilized scattering of the density profiles with depth leading to a similarly stabilized dispersion in the cumulative mass (Fig. 5) which, once divided by the age, yields smaller deviations for older IRHs.

Considering now accumulation rates along IRH1 computed from the DC5 density core (with the error bars on density as proposed in Sect. 5.4) yields results represented on Fig. 8. Also shown is

the same accumulation profile obtained from the S0 density profile and from a combination of all available density profiles along the radar line (according to an inverse distance weighting). Keeping in mind that the resulting errors (black bars on the figure) are probably underestimating the overall




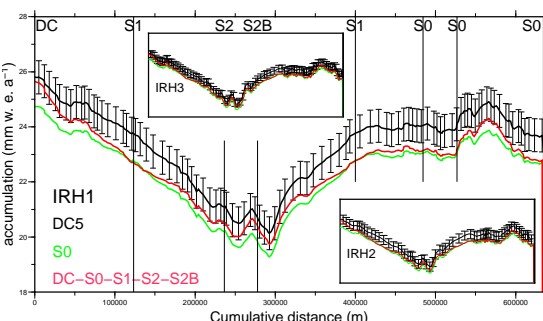

**Figure 8.** Accumulation rates computed from (i) the DC5 density profile (black), (ii) the S0 density profile (green) and (iii) the combination (inverse distance weighting) of all available density cores along the radar line (red). Insets depicts similar representations for IRHs 2 and 3 where the S1,S2 and S2B density profiles have been artificially extended to the required depths by using the S0 fit which offers the best match (compared to DC5) as observable from the insets of Fig. 6.

uncertainty arising from the chosen density profile, and given the fact that similar errors have to be expected with the S0 density profile (not represented for the sake of clarity), one comes to the

conclusion that using the DC or the S0 density does not bring any significant changes in the computed accumulation rates. If globally lower values seem to emerge from the S0 curve with IRH1, the shift rapidly reduces with depth and systematically remains within error bars which become even smaller with IRHs 2 and 3 as visible from the insets. In fact, properly accounting for the geographical distribution of density would normally require integration of all available density profiles evenly

distributed along the radar line according to their respective distances (red profiles on the figure). Similar conclusions emerge in the sense that no significant information is gained, at least for IRH1. Despite an attempt to extend their density profiles down to IRHs 2 and 3, the reduced depth (around 20 m ) of the S1, S2 and S2B cores represent a strong limitation in their actual use. Lastly, given the respective positions of DC and S0 with respect to the radar line, integration of the two density profiles

according to their respective inverse distance would not make much sense (the DC-S0 gradient being perpendicular to the major axis of the radar line). As a consequence, our strategy thus consists in relying on a single but reliable density profile (DC5) rather than trying to integrate the non significant geographical effects from limited or not-so-relevant extra data sets.

**5.7   Overall error budget for surface accumulation rates**

From the expression of snow accumulation rate $\dot{A} = \dfrac{\overline{\rho}\,H}{T}$, the three error sources contributing to the overall uncertainty originate from (i) $\Delta H$ the error in the depth of the IRH, (ii) $\Delta T$ the error in its age, and finally (iii) $\Delta\overline{\rho}$ the error in the depth averaged density $\overline{\rho}$ resulting from the use of the selected density profile. Each of these sources are readily obtained form proper derivation with





respect to the relevant variable of the accumulation rate expression as above and are summarized
in Table 4. For density, assessment of the correct representativeness with a single core was already
derived in Section 5.5 and respectively gives uncertainties of $1$, $0.75$ and $0.5$ mm w.e.yr$^{-1}$ for IRHs
1, 2 and 3. As for the actual errors in the density measurement, we reconsidered the proposed depth
dependent errors of Section 5.4 whose contributions were integrated down to the depth of each of the
IRHs, leading to very similar values around $18$ kg.m$^{-3}$ which once multiplied by $\frac{H}{T}$ yield a fairly
similar depth-decreasing uncertainty. The global error on density is finally obtained as the RMS of
these two contribution as stated in the $5^{th}$ column of the table. For the age, the two independent
terms arising from (i) the relevance and associated uncertainty of the age-depth relationship of the
used core $\Delta T_c$ (8 yr) and from (ii) the positioning error due to radar resolution $\Delta T_p$ (30 yr) give a
quasi systematic RMS remaining around $30$ yr which then induces total uncertainties of respectively
$1.49$, $1.20$ and $0.97$ mm w.e.yr$^{-1}$ for IRHs 1 to 3. Finally, errors in the depth ($H$) of the IRHs stems
from the radar positioning accuracy as determined in Section 4.4 plus a systematic component due
to the time-depth conversion of the two-way travel time. Proper derivation of Eq. 1, with an average
error of $18$ kg.m$^{-3}$ on density gives a relative error of less than 1.5% for the average wave velocity
which results in errors as a function of the IRH depth as in the Table. RMS combination of these two
terms finally leads to the most pronounced error component for the accumulation rate of respectively
$2.54$, $1.68$ and $1.23$ mm w.e.yr$^{-1}$ for IRHs 1, 2 and 3.

**Table 4.** Different source terms in the overall error budget for accumulation rate

| | Depth-averaged density ($\overline{\rho}$) | | | | IRH age (T) | | | | IRH depth (H) | | | |
|---|---|---|---|---|---|---|---|---|---|---|---|---|
| IRH | $\Delta\overline{\rho}_m$ | $\frac{H}{T}\Delta\overline{\rho}_m$ [a] | $\Delta\overline{\rho}_r$ [b] | RMS[a,b] | $\Delta T_c$ [c] | $\Delta T_p$ [d] | RMS[c,d] | $\frac{\overline{\rho}H}{T^2}$ RMS | $\Delta H_v$ [e] | $\Delta H_r$ [f] | RMS[e,f] | $\frac{\overline{\rho}}{T}$ RMS |
| | kg.m$^{-3}$ | kg.m$^2$.yr$^{-1}$ | mmw.e.yr$^{-1}$ | mmw.e.yr$^{-1}$ | yr | yr | yr | mmw.e.yr$^{-1}$ | m | m | m | mmw.e.yr$^{-1}$ |
| 1 | 18 | 1.10 | 1 | 1.49 | 8 | 30 | $\simeq 30$ | 2.97 | 0.22 | 1.5 | 1.52 | 2.54 |
| 2 | 17 | 0.94 | 0.75 | 1.20 | 8 | 30 | $\simeq 30$ | 1.77 | 0.34 | 1.5 | 1.54 | 1.68 |
| 3 | 17 | 0.83 | 0.5 | 0.97 | 8 | 30 | $\simeq 30$ | 1.17 | 0.48 | 1.5 | 1.57 | 1.23 |

(1) Recall that mm w.e.yr$^{-1}$ and kg.m$^{-2}$yr$^{-1}$ are equivalent units

Following (Muller et al., 2010), the cumulative uncertainty is finally proposed as the RMS of
these three main contributors ($\Delta\overline{\rho}$, $\Delta T$ and $\Delta H$) which gives uncertainties of $4.18$, $2.72$ and
$1.96$ mm w.e.yr$^{-1}$ for IRHs 1, 2 and 3 respectively, later rounded to $4.2$, $2.7$ and $2$ mm w.e.yr$^{-1}$ .
The noticeable feature is a strong decrease with IRH depth, in contrast to (Muller et al., 2010) where
an increasing error with depth seems to result from a rapid degradation of the signal-to-noise ratio
due to the weak penetration depth and high sensitivity to water content of their high frequency (2.3
GHz) system. It should be noticed that the proposed uncertainties pertain to the absolute errors on
the accumulation rates and implicitly comprise a significant systematic part that does not come into
play when interpreting spatial or time-dependent gradients as in the following section.



## 6 Accumulation space and time distribution

### 6.1 Spatial distribution

A geographical representation of averaged accumulation rates (and differences between the fields)
over the three periods characteristic of the selected three IRHs is represented in Fig. 9 where signif-
icant trends can be observed. The relatively good phasing between the three IRHs already noticed
on the radargram is observable from the difference in the represented fields (blue scale) not exceed-
ing $1.2 \, \mathrm{mm \, w.e.yr^{-1}}$. It is also consistent with the computed SMB of Fig. 7 and confirms a quasi
stationary accumulation pattern over the past 600 yr. In particular, an overall decrease in accumu-
lation of some 20% from DC to the South West (towards S2) similarly appears for the three IRHs.
More specifically, traversing towards the South East leads to the minimum value between 19 and
$20 \, \mathrm{mm \, w.e.yr^{-1}}$ (depending on the averaging period) at a place close to the S2B core site in the
direction of a nearby megadune field (see Fig. 1).

It should be noted that the 2011/12 TASTE-IDEA traverse was the opportunity for the deployment
of an ultrasonic range sensor (SR50, Campbell scientific) allowing for a continuous monitoring of the
surface height for more than 5 yr at the S2 point. Corresponding surface heights since January 2012
are depicted on figure 10 where short term variations revealing both precipitation and/or strong snow
redistribution events are clearly visible. Despite these rapid changes which eventually contribute to
the meter-scale variability in surface accumulation (Libois et al., 2014), a significant trend emerges
over these 5 yr of measurements. When considering surface densities between $300 \, \mathrm{kg \, m^{-3}}$ and
$330 \, \mathrm{kg \, m^{-3}}$ (Figure 6), this trend maps into accumulation rates between 19.5 and $21.5 \, \mathrm{mm \, w.e.yr^{-1}}$
which appear to be fully consistent with our computed accumulation rate for the S2 point over the
last 234 yr as represented on Fig. 7.

Previous measurements of GPR-derived accumulation rates in the vicinity of Concordia Station
were carried out in 2005 (Urbini et al., 2008). A radar profile through the station revealed an almost
north-to-south gradient of $-0.02 \pm 0.01 \, \mathrm{mm \, w.e.yr^{-1}.km^{-1}}$ in its southern part (see Fig. 6 in Urbini
et al. (2008) and Fig. 11 where corresponding data have been summarized). Despite limited areas
over which they overlap, and a slightly different averaging period (the last 266 yr against our last
234-yr average) these data remain however relatively consistent with ours as can be seen from the
figure. Moreover, from our data, starting from around DC and considering the upper left blue points
of the figure, a difference of about $1 \, \mathrm{mm \, w.e.yr^{-1}}$ can be derived over a distance of some 30 km
leading to a gradient of $-0.033 \, \mathrm{mm \, w.e.yr^{-1}.km^{-1}}$ close to that of Urbini et al. (2008) along a
similar direction. When assessing an uncertainty on our proposed gradient, only the non-systematic
terms have to be considered, such as the effects of a laterally varying density which in the present
case should be insignificant because of the distance of only 30 km. Errors in positioning or age along
IRH1 should also remain small. Indeed, as can be seen from the limited dispersion in both ages and
depths at crossing points S1 and S0 (see Table 3) the spatial variations in positioning or in the age




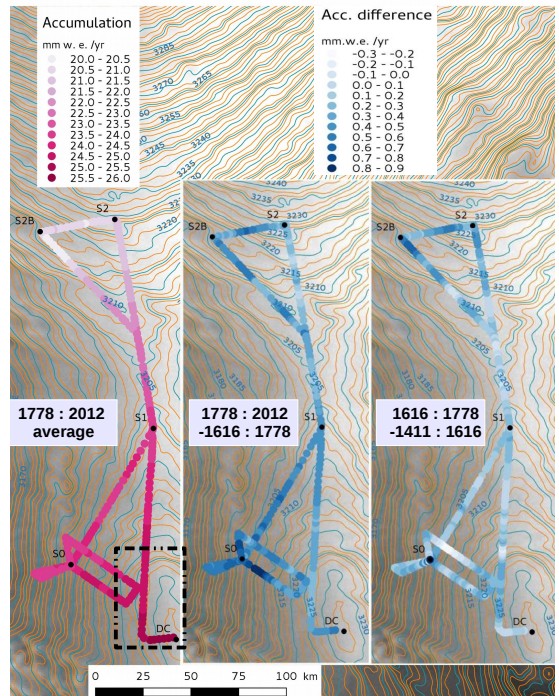

**Figure 9.** Geographical (polar stereographic, $71°S$) distribution of net accumulation rates according to the ages of the three IRHs. The left panel is the average over the last 234 yr (CE 1778-2012) whereas the middle one represent this latter field minus the CE 1616-1778 average. Last, the right panel stands for the difference between the CE 1616-1778 and the CE 1411-1616 periods. Globally positive values in these differences confirm a steady overall increase through time as revealed by Fig. 12. Background is a Radarsat image of the prospected area revealing some topographic features emphasized by the contour line of the surface DEM (Bamber et al., 2009).

along a given IRH are minor (maximum differences of 18 cm and 3 yr for IRH1, leading to a RMS uncertainty of $0.45\,\mathrm{mm\,w.e.yr^{-1}}$ according to the error budget of Table 4. The proposed difference therefore remains significant and leads to an uncertainty of $0.015\,\mathrm{mm\,w.e.yr^{-1}.km^{-1}}$ on the pro-

posed gradient and makes the comparison still relevant. It should be noted that this gradient has also been observed by Genthon et al. (2015) from snow accumulation at two stake networks 50 km apart along a north south direction centred on the Concordia station. Despite a high uncertainty for the uppermost snow density which limits the derivation of absolute values of accumulation rate, the use of a uniform snow density along the 50-km long section revealed a significant gradient similar to

that of Urbini et al. (2008) and of the present study. Their study also shows that despite an overall underestimation of snow accumulation, meteorological analyses from the European Center for





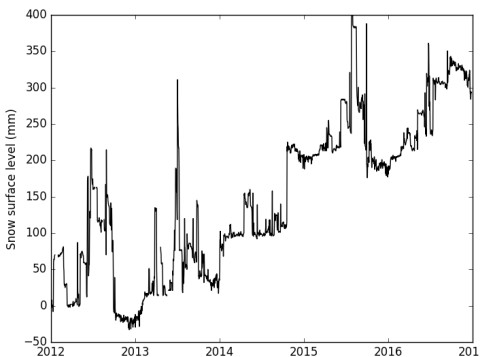

**Figure 10.** Snow surface level in mm above the reference level of the $1^{st}$ of January 2012 obtained from acoustic distance measurements. Presented daily mean values were calculated from measurements recorded every 30 minutes and corrected for air temperature.

Medium-range Weather Forecasts confirm this tendency (see the modeled precipitation field on Fig. 1). Moreover, from a careful observation of precipitation events and analysis of air mass back trajectory pathways Genthon et al. (2015) show that the core of snow fall results from relatively warm (and hence humid) air masses intrusions from the north which undergo a significant temperature-driven depletion in moisture when passing over the Concordia dome on their way to the south. This is fully consistent with the observed gradients.

## 6.2 Time-dependency of accumulation

Because of the limited number of IRHs being analysed in this study, only a coarse characterization of the time-dependent accumulation can be proposed as is the case in (Urbini et al., 2008) for instance. This limitation comes from the stringent requirements for a reflector to be considered as properly isochronous (showing the continuity over large distances) and assigned a relevant age (by passing through a coring site where an depth/age relationship can be independently derived). Focusing on a given location and considering a large number of reflectors might be tempting at first sight but this is only possible if every reflector is unambiguously connected to an ice core in order to account for the unavoidable spatial changes in the depth distribution of IRHs. As for recent accumulation rates, it should be noticed that the multiple direct reflections in the air between the emitter and receiver parts of the antenna leads to a saturated signal screening potential sub-surface reflectors. The last decades of surface mass balance from GPR are therefore often missing which prevents proper comparisons with 'modern' accumulation values as derived from stakes farms for example (only operational since the late nineties for the DC area).





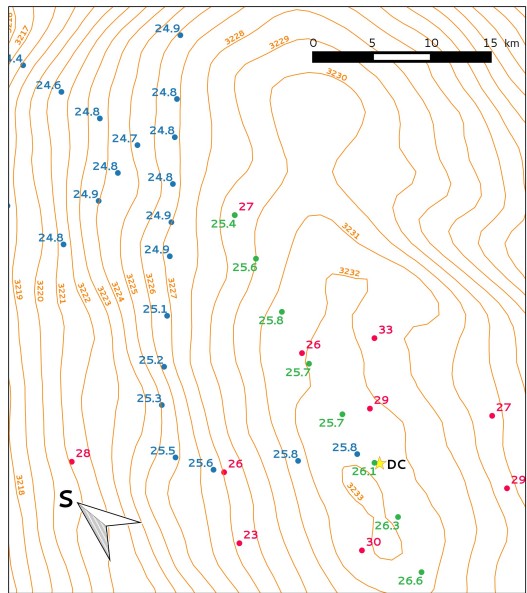

**Figure 11.** Comparison of our accumulation rates with those in (Urbini et al., 2008) in the vicinity of Concordia Station over the area featured by the dashed inset of Fig. 9. Blue dots represent the rates as represented in Fig. 9 for the last 234 yr whereas the green ones are those from the radar transect of (Urbini et al., 2008) after being equally sampled (5 km) and averaged over five measuring points. Also shown by red dots are accumulation rates inferred from an analysis of firn cores based on tritium/ $\beta$ markers for the 1965-2000 period (Urbini et al., 2008, an references therein) and for which an uncertainty of $10\%$ has been derived. Contours are altitude a.s.l. as given by (Bamber et al., 2009).The Concordia Station is featured with a yellow star.

We therefore propose three periods over which mean accumulation rates can be derived (1411-1616, 1616-1778 and 1778-2012 CE) as can be seen from Fig. 12. Data were spatially averaged over five main sectors whose respective extensions are reported on the figure with appropriate symbols.

The results first confirm the stationary character in the spatial pattern of accumulation rates as was already found from inspection of Figs. 7 and 9 for example. Indeed, the trend in the time-dependent increase over the three proposed periods is remarkably similar from one sector to the other. Similarly to the above mentioned spatial patterns, only the non-systematic parts in the associated errors (with regard to time) have to be considered for interpreting these time trends. Therefore, the potential

deviation in the isochronous character of the IRHs around 10 to 15 yr, of a similar order as the age dispersion of the IRHs at crossing points as given in section 4.4 essentially comes to play. It leads to a potential margin of error ranging from $1.07$ to $0.42\,\mathrm{mm\,w.e.yr^{-1}}$ for respectively IRHs 1 to 3 which would theoretically hamper the significance of the computed trends. However, the fact that these data represent averages over entire sectors and the remarkably similar evolution from one



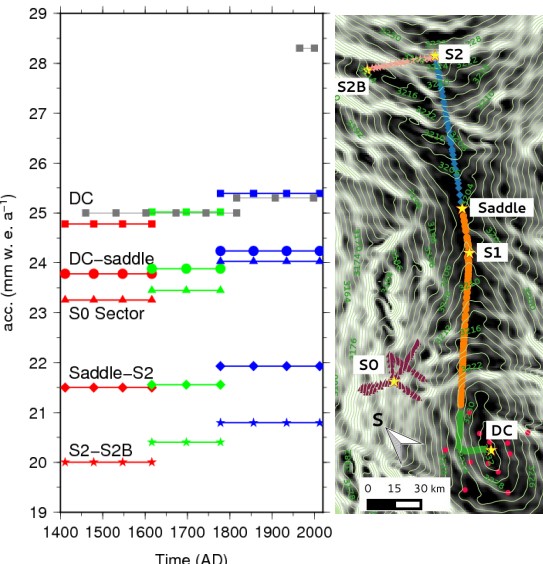

**Figure 12.** Time-dependent evolution of the accumulation rates spatially averaged over five sectors along the radar profile. The first sector corresponds to the DC vicinity (green squares on the map) within the firn core network as in (Urbini et al., 2008) and featured by the red spots on Fig. 11. The second sector more or less follows the ridge from the end of the DC sector down to a topographic saddle (orange circles). The third (blue diamonds) continues along the ridge and climbs up to the S2 point. The fourth (purple stars) follows the perpendicular S2-S2B transect whereas the fifth (brown triangles) comprises the vicinity of the S0 point. The time frames for the time-averaging (left of the figure) correspond to the age difference from each of the three IRHs to the next above or to the surface, namely 1415-1616, 1616-1778 and 1778-2012 CE. Also presented with grey symbols are the accumulation rates as reported by Frezzotti et al. (2005) and (Urbini et al., 2008) over the DC area (see text). Surface topography with 2-m contours is that of (Bamber et al., 2009).

sector to the other gives some credit and suggest a regularly increasing accumulation rate for this sector of the Antarctic plateau over the last 600 yr.

Our results also compare well with those mentioned in (Urbini et al., 2008) and references therein over the DC area (depicted in grey in the figure). More specifically, from $nssSO_4^{2-}$ volcanic spikes along the EPICA EDC96 core, an average accumulation rate of slightly less than $25 \, \mathrm{mm \, w.e.yr^{-1}}$

for the 1460 (Kuwae)-1816 (Tambora) period was proposed (Castellano et al., 2004). Then follows the 1816-1998 AD period with a $25.3 \, \mathrm{mm \, w.e.yr^{-1}}$ accumulation rate followed by a marked increase up to $28.3 \, \mathrm{mm \, w.e.yr^{-1}}$ between 1965 and 2000 deduced from nuclear test horizons along the firn cores represented by the red circles on the figure (Frezzotti et al., 2005).

Results from farm stakes are also proposed by Frezzotti et al. (2005) with $32 \, \mathrm{mmw.e.yr^{-1}}$ for

the 2004-07 period and up to $39 \, \mathrm{mm \, w.e.yr^{-1}}$ between 1996 and 1999, but their relevance is ques-



tionable because (i) the measuring period is extremely short with respect to a very high inter-annual
variability (ii) the derived accumulation rate requires proper knowledge of the sub-surface density
which is highly variable and difficult to accurately measure and (iii) the intrinsically strong spatial
variability in snow thickness at the scale of a farm stake also requires long integration periods to
gain in significance. This is confirmed by the associated standard deviation of $14$ mmw.e.yr$^{-1}$ for
the 1996-99 stake results. In any case, no comparison with radar data is possible because of corre-
sponding reflectors too close to the upper surface.

### 6.3   Interpretation in the frame of remote studied areas

Several past studies (e.g., Fujita et al., 2011; Frezzotti et al., 2004, 2005) have proposed explanations
for the spatial distribution of the accumulation rate on the Antarctic plateau as a function of the envi-
ronment. They globally come to the same conclusion according to which surface mass balance results
from a large scale synoptic precipitation pattern and generally smaller scale ablation/redistribution
processes mainly driven by the combination of the wind and the surface slope (SPWD, Slope along
the Prevailing Wind Direction). From data in the Dome F-Dronning Maud Land sector, Fujita et al.
(2011) show that a significant amount of the total precipitation deep inland comes from the synoptic
scale advection of moisture-laden and warm air masses from low latitude usually characterized by
rather strong associated winds as was also demonstrated by (Genthon et al., 2015) for the DC area.
Most of the moist is released during the orographic climb towards the plateau or on the windward
side of any major ridge. Conversely, in the case of a ridge, on the leeward side, an adiabatic warming
of the descending air masses reduces the condensation potential even more leading to the concept
of 'rain shadow'. According to the major air mass trajectory pathways as described in figure 5 of
Genthon et al. (2015) and more specifically the three paths from the east accounting for $55\%$ of the
total precipitation, a variation in the large scale precipitation field is expected across the Vostok-DC
ridge as confirmed by our data, at least in the vicinity of DC and towards S0. As for the large scale
gradient along the radar line, a similar analysis would require a detailed picture of the circulation
pattern which is beyond the scope of this paper. Still, a continental effect as noticed by (Fujita et al.,
2011) in the Dome Fuji Dronning Maud Land sector seems plausible and is in line with the com-
monly accepted accumulation gradient along the ridge between DC and Vostok. Finally, the lowest
measured accumulation values around the S2B point are clearly situated off divide. Although it is
difficult to assert whether they result from a corresponding low in the synoptic accumulation pat-
tern or from smaller scale redistribution, the less regular surface topography probably contributes to
more local erosion/deposition processes resulting in more pronounced gradients than along the main
divide.

Again, comparing the time dependency of our results with those from other studies suffers from
the lack of correspondence between the averaged periods. Osipov et al. (2014) and Fujita et al.
(2011) give a detailed overview of the time evolution of the accumulation rate over several sites on





the Antarctic plateau during the last millennium. More specifically, a correlation between climate
and accumulation changes seems to emerge from most studies cited in (Osipov et al., 2014) and ac-
cording to which the slightly decreased temperatures during the Little Ice Age led to a corresponding
decrease in accumulation rate. However, not only is the timing significantly different from one lo-
cation to the other, but some places have undergone inverse trends such as Siple or Vostok where,
for example, the CE 1261-1601 lower accumulation period (-12% compared to the CE 1260-2010
average) was followed by a 13% increase between CE 1661 and 1815. The authors attribute this
positive anomaly to a high sensitivity of the large-scale circulation pattern which undergoes shifts
in the stream pathways. Strong latitudinal gradients in the accumulation rate in the Vostok area are
confirmed by Ekaykin et al. (2012) who stresses this sensitivity but also invokes possibly strong
snow redistribution along the Dome B-Vostok ridge due to the combination of wind and surface
slope. The time dependency of our data would have matched that of Vostok should our intermediary
period (CE 1616-1778) have undergone a similar anomaly. Given the sensitivity to these low accu-
mulation rate areas to the above mentioned processes, this partial mismatch probably illustrates the
stochastic aspect of the accumulation changes there and does not constitute an inconsistency. Last,
a recent positive trend in accumulation over the last decades and more generally since the middle
of the 19th century emerges from most studies on different locations on the Antarctic plateau (e.g.,
Mosley-Thompson et al., 1993; Fujita et al., 2011; Frezzotti et al., 2013) and especially in the DC
area (see Sect. 6.1). Although they result from measurements over short periods (for instance from
the Pinatubo volcanic horizon or from recent stake farms) and are therefore suffering larger uncer-
tainties, they remain compatible with the recent observed warming in most places in Antarctica. If
such a recent trend cannot be observed from our data (see Sect. 6.2), it can still be implicitly included
in our last 234-yr average and contribute to the associated increase in accumulation rate.

**7 Conclusion**

Relative depths of three IRHs from a 630-km long GPR profile have been combined with time
markers as well as density profiles along ice core data to provide surface accumulation estimates
between the DC and Vostok stations on the Antarctic plateau. Results show a remarkably persis-
tent accumulation pattern, whatever the investigated time period (1411-1616, 1616-1778 and 1778-
2012 CE) during the 600-yr total coverage. More specifically, a significant NE-SW gradient from
$26 \pm 2$ mm w.e.yr$^{-1}$ at DC to $19 \pm 2$ mm w.e.yr$^{-1}$ at the other extremity of the profile emerges
and appears to be consistent with previous radar data available over the 25 km from DC. As for the
time dependency, a steady increase of about 5% is also detectable over the last 600 yr and partially
matches that from similar radar data. A careful error analysis is proposed that accounts for all possi-
ble intervening terms and provides depth dependent margins of error from 4 mm w.e.yr$^{-1}$ (234-yr
average) to 2 mm w.e.yr$^{-1}$ (592-yr average). It also shows that despite the proven isochronous



character of the proposed IRHs, the main source of error is to be found in the uncertainty in the determination the IRH depth which maps onto an age uncertainty of some 30 yr which eventually accounts for more than half of the total uncertainty on the accumulation rate. The error budget also

shows that in our case, the uncertainty in terms of density resulting from both the representativeness and measurement errors of a single core is of the same order as the changes expected from incorporating the potential geographical variability in density from extra (but less reliable) cores along the radar line. In other words, it proved better to exclusively rely on a single reliable and accurate density core at one extremity of the profile, rather than trying to incorporate doubtful spatial changes

from less reliable or even non exploitable intermediate density cores.

Measuring surface mass balance over Antarctica remains a challenge and amongst the different available methods, combined radar and ice core data provide a robust means of properly assessing large scale spatial patterns as well as long-term temporal changes of snow accumulation. This is fundamental for addressing the overall mass budget of the ice sheet, especially in the context of global

warming when increased accumulation from more moisture-laden ocean air masses compete with enhanced ice flow through outlet glaciers which casts some doubts on the future contribution of the ice sheet to future sea level. Knowing surface mass balance space and time distribution is also fundamental for interpreting and dating the ice core climatic signal. In this respect, the DC-Vostok area is of prime interest for the retrieval of long climate records by combining large ice thicknesses and low

accumulation rates. This area therefore becomes the focus for the quest of new coring sites where an ice archive potentially older than a million year could be exploited (project 'Beyond EPICA Oldest Ice'). Surface mass balance maps thus constitute a major input for selecting the coring site (e.g., Fischer et al., 2013). However, despite the large possibilities of the proposed method for providing large scale accumulation fields, a comprehensive and high resolution coverage of the entire Antarctic

ice sheet is not realistic. Surface mass balance results from a subtle interplay between the regional accumulation pattern and more local parameters such as the surface topography and the wind field. Outputs from global circulation models associated with the local specific environment should allow for relevant surface mass balance computations. This requires deriving accurate parameterizations describing the influence of the association of surface topography and the wind field (such as the

Slope along the Prevailing Wind Direction, SPWD, see for example (Frezzotti et al., 2007)). Radar data as proposed in this study are intended to be used for constraining/validating such relationships leading to a forthcoming paper.

*Acknowledgements.* We would like to thank the French Research National Agency (Project ANR-07-VULN-013 VANISH) who provided a significant financial support to the TASTE-IDEA 2010-11 traverse from which

data of the present study originate. The French Polar Institute (IPEV) was crucial in the success of this traverse by mastering all the logistical aspects, providing the necessary staff, material and consumables. Technical staff (IGE: E. Lefebvre, G. Teste and IPEV: A. Vende, A. Leluc, D. Colin) is also heartily thanked for its invalu-



able expertise. The Russian Arctic and Antarctic Expeditions (AARI, Vostok base) are also thanked by having welcomed the traverse at the Vostok station. Project ANR-06-VULN-016-01'DACOTA' also contributed by

funding part of the GPR used in the present study.



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
