# Peer review of "Spatial and temporal distributions of surface mass balance between Concordia and Vostok stations, Antarctica from combined radar and ice core data : First results and detailed error analysis."

_The Cryosphere, 2017_

## Referee Comment (RC1) · E. Isaksson (Referee) · 16 Aug 2017

This paper concerns new data regarding surface mass balance from interior East Antarctica, collected during IPY as part of the project TASTE-IDEA. Obtaining reliable data of surface mass balance from these low accumulation areas is very difficult and this paper takes on this challenge in a careful and systematic way. The authors have

used a combination of radar (GPR) and shallow ice cores and have thus been able to achieve a good coverage of both spatial and temporal distributions of SMB during the past 600 years. The fact that this area is a possible location for a new deep ice core (Beyond EPICA-Oldest Ice project) makes this paper particularly important and interesting. It is very important to have field data for validation of models and then it is absolutely crucial that the field data are of good quality. This paper is an excellent example of such a study! It was a true pleasure to read this paper; it is exceptionally well-written, has a good structure and the data is well presented in both figures and text. In particular I like the careful consideration of the density and error analysis. In my view this paper is more or less ready to be accepted. I can only point out some editorial issues to correct- I recommend the authors to proof read the paper a little more carefully.

Editorial comments

Be consistent with the use of Fig. , Figure, Figs throughout the paper.

Examples:

o. 11.line 278 "colored dots on the figure"- which one?

p.11 line 285. "Figs1 and 2"

p. 14, line 336. Which figure?

p. 14 line 356 "Figure 4" ,

p.11 line 285. "Figs1 and 2"

- and many more.

The same issue also goes for the use of Section, Sect.

p. 9 lines 243-248. Not consistent use of CE and AD. There are many more examples in the paper so please go over and change these.

p. 14, line337. "Volcano" should all be lower case.

p. 15, line 368. "Sweden" is misspelled

p. 20. Line 498. Section ?

p. 24, line 587.should be: Following Muller et al. (2010), the same problem with parenthesis on line 590.

p. 27. Line 661. Should be: late 1990ies

p. 28. Figure caption first line: should be Urbini et al. (2008), same issue for line 3.

p. 29. Figure caption: same issues with parenthesis for references as in previous caption - and actually many similar issues throughout the paper. I found particularly many in sections 6.2 and 6.3.

---

## Author Comment (AC1) · 18 Aug 2017

We are paticularly grateful to reviewer#1 for her very positive review.

I admit some slack in the proper use of abbrevations and in the respect of writing conventions.

We will wait for the forthcoming reviews and then do the required corrections of the

paper

E. Le Meur , in the name of all co-authors

---

## Referee Comment (RC2) · Anonymous Referee #2 · 8 Sep 2017

Review of Le Meur et al. 2017: Spatial and temporal distributions of surface mass balance between Concordia and Vostok stations, Antarctica from combined radar and ice core data: First results and detailed error analysis

Overview: Le Meur et al. present GPR profiles and limited ice-core data to determine the spatial pattern of accumulation near Dome C, East Antarctica. The work is motivated by the effort to find a location to drill an ice core to recover a >1 million year

continuous climate record. An accurate surface mass balance is prerequisite for site selection. Thus, the primary conclusion of lower accumulation rates towards Vostok is important, if not particularly surprising.

My impression of this manuscript could not be any more opposite than the first referee. I found this manuscript tedious, poorly written, and ill-considered. While the main result is of interest, this manuscript suffers from a number of flaws; in particular, a lack of brevity. I found myself asking the same question over and over: why does this take ∼13,000 words?

I want to highlight two specific area where I fundamentally disagree with the other reviewer. First, to address the other reviewer's comment: "In particular I like the careful consideration of the density and error analysis". I found the density and error analysis to be lengthy, but nor informative. For instance, error bars are shown on the accumulation rate inference using the DC5 density profile in Figure 8. Yet they don't overlap the accumulation inference using the S0 density profile. When I read the caption of Figure 8, it does not even mention the error bars. When I then track down the reference in the text to Fig. 8, I am referred to section 5.4. Once there, I cannot even find what uncertainty the authors are referring to. There is this section of text:

"As a consequence, an uncertainty of $\pm 20$ kg.m$-3$ over the first 7 m linearly decreasing to $\pm 15$ kg.m$-3$ at 15 m deep and remaining constant further down is proposed for these density measurements (see error bars on the figure)"

But this doesn't directly translate to the error bars in Figure 8. The authors go on to write this about Figure 8:

"Keeping in mind that the resulting errors (black bars on the figure) are probably under-estimating the overall uncertainty arising from the chosen density profile, and given the fact that similar errors have to be expected with the S0 density profile (not represented for the sake of clarity), one comes to the conclusion that using the DC or the S0 density does not bring any significant changes in the computed accumulation rates"

So let me get this straight. First you say the errors are an underestimate. Second you say the S0 density profile is not a lower bound because it is also uncertain. Then third you jump to the conclusion that the density profile is not a significant source of uncertainty. This flabbergasts me for two reasons: 1) The uncertainty on the inferred accumulation rate (based only on density and not considering other factors) is then somewhere between 1 mm/yr (roughly the difference between S0 and DC5) and 2 mm/yr (roughly the difference between the upper error of DC5 and lower error of S0) which is a third of the total variation in your 600km line survey. This is not insignificant. 2) But if the impact of the density profiles truly was "insignificant" then why did you spend so much space writing about?

Therefore, I don't find the error analysis "careful"; I find that the authors are obscuring clarity with unnecessary detail.

Second, I want to address the dating of the S cores. It becomes clear when the authors admit they can't distinguish Tambora from Cosiguina that the dating for the S cores is not reliable. Sulfate leaves no chemical composition information to tie events in cores together unlike tephra. Therefore, you have to match events based on patterns or amount of sulphate deposition. Tambora and 1809 are the closest thing in the last 500 years to a specific pattern match. I've attached a figure of sulfur from WAIS Divide (because it is the best resolved ice core in Antarctica, Sigl et al., 2013) to illustrate just what the authors mis-identified. I realize that the amount of deposition may vary by location in Antarctica, but if you cannot reliably distinguish the couplet of Tambora and 1809 regardless of magnitude, what are you actually matching? The authors note that Tambora was missing is two of the five Volsol cores, but there is nothing to suggest that Tambora itself is prone to being missed, only that small-scale irregularities between replicate cores can obscure volcanic events. Thus, Cosiguina is as likely to be missing as Tambora. It is particularly odd that this manuscript finds more events, and ties these events to specific volcanoes, than do the 5 replicate Volsol cores (see Gautier et al., 2016 Table 1).
But this brings up another point, the authors are not presenting any of the sulphate data anyway. I'm expected to accept your matches on blind faith. All of this would not be nearly as annoying if it weren't for the fact the manuscript could be written entirely without any dating of the S cores. The DC (Volsol) cores provide a robust age scale. The IRHs are accepted as isochrones without the cross-over dates (which in theory are useful, but would require reliable dating which you don't have). So why bother with this at all?

Overall, I find this paper to be unreadable. Not because it is outright wrong, but because there is so much extraneous information that the important points are lost in a sea of detail. Frustratingly in a paper of this length, there is little in the way of discussion of the implications of these results. How about a few paragraphs on what this suggests for where to drill a deep ice core? How about comparing the 5% increase in accumulation to inferred air temperatures? Is it consistent with Clausius-Clapeyron? And what is the impact on global sea level? Over what spatial area would the increase have to be to have a noticeable effect on sea level? And a figure would really help in the comparison to accumulation inferred from other ice core sites.

I have not provided specific comments to the text for a simple reason – I believe the vast majority of the text should be deleted. The figures are in better shape, but the paper really only needs three: Figure 1 Figure 2 combined with Figure 7 and full uncertainties (not those shown in Figure 8) Figure 9 combined with Figure 12

I challenge the authors to write the manuscript in 5000 words or less. There are good thoughts within the manuscript but they are obscured by unnecessary detail and lead to incomplete thoughts on some of the important aspects. The main result of a spatial map of accumulation and an increasing trend through time is useful and should be published. The manuscript in its current form should not.
* * *
[Figure]

Fig. 1.

---

## Author Response (AR1)

Reply to reviewers comments on paper entitled '**Spatial and temporal distributions of surface mass balance between Concordia and Vostok stations, Antarctica from combined radar and ice core data : First results and detailed error analysis.**' by Le Meur et al.

Dear Editor,

Please find below our reply to the 2 reviewer's comments on the above cited manuscript.

Whising you good reception,

Sincerely,

Emmanuel Le Meur in the name of the authors,

**Reply to reviewer 1** :**

Review form reviewer 1 was very positive in the sense that only editorial comments were proposed. For parts of text that were kept, the proposed changes have been done.

**Reply to reviewer 2** :**

**General reply**

The main point raised by the reviewer is a lengthy paper with too many details that makes the reading tedious and obscures the message to deliver.

We acknowledge that it is partly true. One explanation resides in many different subjects tackled (core chemistry, radar interpretation, error analysis... ) implying different contributions from as many authors aiming at seeing their work fully described. This is particularly true for the inclusion of results from the S0 core. We also agree on the fact that the error analysis as initially proposed was unclear and also contained minor conceptual errors.

We tried our best to answer these remarks by reducing the text and figures and by partly rewriting portions of text (see details below). Parts of text and figures have also been moved to the appendices so that less important issues do not obscure the reading. However, reducing the paper to 5 000 words or so is in our view too drastic given the wealth of data and the fact that, amongst papers dealing with radar results, this draft is, to our knowledge, the first one presenting a rigorous and comprehensive (at least now) treatment of the budget error for radar-derived accumulation rates. In its 2-column version, the new paper (without appendices) is 4 pages less (for a total of 15 pages) and more importantly has been rewritten for a more fluent reading.

Reducing the figures to the extent suggested by Reviewer2 for the sake of brevity is not reasonable given their information content which we believe is necessary for the comprehensiveness of the paper. Here follows a summary of how we dealt with figures :

> - Removing Fig. 3 is unrealistic since the dating process against an ice core is a key process in the proper derivation of accumulation rates. However, because they are of less importance, plots of the S0, S1, S2 cores have been moved to Appendix C where they serve for estimating cross errors for the IRHs dates. Age/depth relationships for these cores

are probably less reliable than that of the reference VOLSOL-1 core, but they are still reliable enough for estimating the age dispersion as is done in the appendix and which serves later for the global error budget. Accordingly, the corresponding table giving the ages dispersion has also been moved to the appendix.

- Fig. 4 has also been moved (to the Appendix B). It is necessary for explaining the representativeness of a single core for a place like DC. Here, only the result really matters and is solely given in the main text. The reader is thus invited to refer to Appendix B to apprehend the age dispersion of the VOLSOL cores leading to the dispersion figures proposed in the main text.

- Fig. 5 was kept since it is essential for the representativeness of a single core in terms of density this time (also necessary for the final error budget). Corresponding text was also partly modified but mainly kept because it often refers to the figure which therefore justifies the presence of the 2 in the main frame rather than in an appendix.

- The top of figure 6 (cores S1,S2, S2B) has been removed because of the limited interest of the represented S1,S2, S2B cores in terms of density content. However, the bottom representing the 2 Explore and DC5 density cores has been kept and merged on top of Fig.5. Given the importance of density, it seemed unreasonable not to present a density curve in the paper (as suggested by the reviewer). Reviewer 2 argue that the S0-Explore core should be disregarded in the paper. We do not believe so. We agree on the fact that the interpretation of the 10.48-m horizon is problematic in the S0-Explore core. It was therefore agreed to only consider the deepest reflectors of the Explore core to supplement the first 20 m of the S0 core so as to provide a hybrid S0-Explore age/ relationship for the S0 site (explained in the Appendix C). We believe this relation is still reliable enough to be used for estimating the age dispersion as is done in the appendix (and depicted in the companion figure in the appendix). Corresponding values are then considered in the main text. Last, density measurements along the Explore core are fully exploitable and are necessary for conducting the sensitivity test on accumulation rates to the different geographical cores from which one essential conclusion of the paper is drawn. Of course the text has been reconsidered so as to account for these changes and make the remaining main text consistent and fluent.

- Fig. 8 has been removed with its content in terms of error bars reported in fig.7

- As for the spatial and time dependent accumulation rates, they constitute the core of the results of the paper and their respective importances led us to expressly treat them in 2 separate paragraphs and hence in 2 figures, contrary to the suggestion of the reviewer consisting of merging Figs. 9 and 12. Moreover, merging the 2 figures in one would not bring any significant gain of space and would, in our view, alter the fluidity in the reading.

- Reviewer 2 also implicitly suggests to remove Figs. 10 and 11, but we feel they are worth presenting by providing extra independent data sets against which to compare our results.

Finally, the main frame of the text has been reduced by almost 4 (double column) pages and now includes 9 figures instead of the initial 12. Given the opposition between the 2 reviewers, we feel that most suggestions of reviewer 2 in order to shorten the paper are too drastic and that the paper can sufficiently gain in conciseness from an intermediate response as proposed here.

**More specific comments and corresponding response from the authors** :

**- With regards to error bars of Fig. 8 and their interpretation** :

Error bars on Fig. 8 in the initial version were representing the lower and upper bounds for accumulation rates computed when using the upper and lower density profiles of Fig. 6 when considering respectively the top and the bottom of the depicted error bars on the figure. These density error bars were those initially described in section 5.4 ("+- 20 kg/m3 for the first 7 m linearly decreasing to +-15 kg/m3 at 15 m deep..." ). After reconsideration of the paper, it appeared that not only the text (and associated figure) is totally ambiguous but the reasoning is also dubious.

This is why we now propose an alternative method in which we integrate both the density measurement errors and the uncertainty arising form the representativeness of a single core into a single term derived from the dispersion in the accumulation rates computed from all the reliable density cores drilled at DC. This is now explained at the beginning of section 5.5, more specifically with the sentence : " Because they result from different projects implying different measurements protocols, the numerous cores drilled in the DC area provide a means of estimating the order of magnitude of the combined effect of these two terms ". This resulting combined error term is therefore estimated form the RSM dispersion of the different accumulation rates issued from these various cores obtained by summing the masses down to the require depths as shown in the new Fig. 4.
This combined error term (represented by the error bars of the insets of the new Fig. 5) is later compared to the changes in accumulation rates now obtained from using either the DC5 reference density profile or the S0-Explore core, in order to assess the sensitivity to regional changes in density. From the overlap of the error bars on Fig. 5 we come to the conclusion that accounting from the geographical changes in density at our disposal does not bring significant changes to the resulting computed accumulation rates as explained in the new Sect. 5.5.
The uncertainty solely arising form measurement errors along the DC5 core ( +- 20 kg/m3 for the first 7 m linearly decreasing to +-15 kg/m3 at 15 m deep... ) will only serve the purpose of estimating the uncertainty in the inference of the vertical velocity profile later used in the overall budget error section (Sect 5.6 and corresponding Table 3 that were consequently reconsidered). One extra reason for this choice is the rather arbitrary character of these density measurement errors (a very difficult term to assess). These (potentially questionable) error terms will anyhow have a minor impact as can be seen from the small contribution of the uncertainty in the velocity wave in the overall budget error compared to other terms (see term $\Delta H_v$ in column 10 of Table-3).

As a result, the entire Section 5 has been reconsidered, shortened by now including 6 subsections (instead of the initial 7). It did not drastically change our conclusions, notably the unnecessary consideration of the geographical changes along the radar profile (at least in the current context). However the question was worth asking and answering as rigorously as possible since it leads to the important conclusion that before considering the potential regional changes in density, an accurate error estimation due to the local density error should be considered. This new approach led to slightly different figures in the global error budget as represented in Table-3, but the final result is only slightly changed in terms of total maximum margins of error.

**- Problems raised by the S0 core**

As for the problematic dating of the S0 core, we agree that it is certainly less reliable than that of the reference core (VOLSOL-1). This is the reason why the TASTE-IDA cores (including the S0-Explore core) are not used for dating the IRHs (we exclusively use the reliable VOLSOL-1 core for that).

However, age/depth results for these cores result from a careful analysis protocol relying on detection of both volcanoes and radio-isotope horizons. Moreover, the ambiguous 10.48-m deep horizon of the Explore core is not accounted for in the derivation of its age/depth relationship. This latter is derived from horizons from the S0 core supplemented by only the deepest (> 20 m) ones of the Explore core. No shift at the transition is observable and the resulting quadratic fit still exhibits a r2 greater than 0.99.

It should be noted that these extra cores are exclusively used for estimating the age dispersion along the selected IRHs in order to test their isochronicity. Corresponding text and figure were then moved to the Appendix instead of being simply removed because they are our only means of validating the radar approach. We insist on the fact that this paper is one of the first where a big effort is put on trying to validate the isochronicity of the IRHs and propose an as rigorous as possible error budget on the resulting accumulation rates.

Although we do not strictly follow the reviewer suggestion, text and figures devoted to these cores has been significantly reduced in the main frame of the paper. The text was also modified in order to make this new approach clearer as described below :

- Last paragraph of the introduction : We are not anymore talking about '4 age depth relationships later used for dating purposes '. Instead we say 'Ice core data are then presented with a focus on the depth markers obtained from chemistry analysis (volcanoes) and/or radio-isotope counting (nuclear tests) leading to the VOLSOL-1 age-depth relationship later used for dating purposes.'

- Beginning of section 3 : a small paragraph has been added in replacement of the old 'Ice core drilling ' section in which we clearly say that only the VOLSOL-1 core is used to time calibrate the IRHs and that the less reliable cores (S1 S2 and S0-expl) only serve as cross-over points for assessing estimates of the age dispersion along our IRHs.

- Beginning of section 4.2 (Methodology for dating the IRHs): The reasons for the above choice are precised and the way the hybrid S0-Explore age/depth relationship has been derived  is precised with reference to the corresponding figure in Appendix C.

- End of section 4.3 : From the dispersion results of the Appendix C, the isochronous character of our IRHs is presented.

- Last, end of Appendix C : We explain why the results of these cores should be considered with caution, but on the other hand why we think these results are still exploitable for estimating the age dispersion.

**-With regard to the last suggestions of Reviewer 2 consisting of adding a few more paragraphs**

In a process of shortening the paper, we found it difficult to address most of the extra issues proposed by Reviewer 2. Apart from one (impact in terms of sea level), we did not feel like addressing them for the following reasons :

- 'How about a few  paragraphs on what this suggests for where to drill a deep ice core ?'

→ Low accumulation rates are a prerequisite for expecting long climatic exploitable records form ice cores ... but it is one necessary condition among many other ones like (i) a large ice thickness, (ii)  specific temperature conditions at the base of the ice sheet partly driven by the geothermal heat

flux, (iii) rather smooth bedrock topography for preventing disruption in the stacking of basal layers, etc... Most of these issues are beyond the scope of the paper. In the introduction we mentioned the interest of low accumulation zones for the  quest of old ice as a necessary condition, which we only present as one justification (among others ) to study surface mass balance on the Antarctic plateau. We do not see what else could be said in the framework of a paper dedicated to a methodology for measuring surface mass balance measurements.

> - 'How about comparing the 5% increase in accumulation rates to inferred temperatures ' Is it consistent with Clausius Clapeyron ?'

→ To our knowledge, there is no reliable temperature curve over the last 200 yrs on the Antarctic plateau, not even over the past 50 yrs or so. Models usually fail to reproduce temperature evolution over places where the lack of data (in terms of spatial coverage and duration) prevents data assimilation technique. It often leads to biases requiring correcting factors like in the ERA Interim model for instance. Moreover, air masses in these area are prone to intense supersaturation (Genthon et al., 2017,ACP) that precludes the existence of a physically consistent relationship to Clapeyron. As demonstrated by Genthon et al., (2015) and also partly mentioned in the text, explanations for the modern evolution of the SMB around DC is more to be found in changes in the storm track pathways.

> - 'And what is the  impact on global se level ?'

→ As for the suggested impact on sea level, the end of Section 6.3 has been supplemented by a couple of sentences that help to translate our average accumulation increase (as deduced from Fig. 9) into equivalent sea level contribution, should the observed increase apply over most of the Antarctic ice sheet. It also allows a comparison with results of the simulations of Krinner et al., (2006) about the possible contribution of Antarctica in a warmer climate by the end of the 21st Century as mentioned in our introduction.

**-With regard to the very last suggestions of Reviewer 2 concerning adding a figure to compare with resuls from other sites :**

→ As for the last suggestion of Reviewer 2, we indeed initially thought of a figure  synthesizing results from  other sites, but it appeared practically difficult to do since the available examples gives general tendencies but no real figures to put in a graph. Again the problem of different averaging periods over which the proposed trends apply makes it difficult to put corresponding results in a relevant figure. Last, the proposed examples are often tendencies as a result of a specific topographical or geographical context (position with regard to a ridge, continental character...) which can not be properly represented in a synthetic graph.

[revised manuscript text omitted]

---

## Author Response (AR2)

Dear Editor

Please find attached our revised manuscript following the comments from the third reviewer.

5     - Rev Comment 2 : "The biggest obstacle to inferring temporal accumulation rate variability at these low rates (21-25mm w.e/yr) is the underdetermined firn density profiles along the radar route. Hence, I think that the section 6.2 Time-dependency of accumulation is overly speculative. The temporal rate of change is either at the limit of detectability or below it, depending upon the cumulative errors in its determination."

10     -> We agree on the speculative aspect of our derived time trend. In that respect, we reconsidered the error budget pertaining to these time trends by removing the time-independent biases (as explained in the new text) and most importantly, added corresponding error bars on figure 9 which now makes much more sense. It becomes now clearer that it is difficult to unambiguously derive significant trends. As a consequence, all text dealing with the time dependency has been rewritten

15 accordingly (abstract, introduction, section 6.2 and conclusion). Section 6.2 is maintained (recall that reviewer 3 did not ask for it to be deleted) but significantly reconsidered according to the new version of the figure and the fact that we have to be more cautious in the interpretation. We kept some references to previous work in the area to show that our results remain compatible within the limitations imposed by the error bars. Although a bit frustrating, this remains an information that we

20 feel it is important to pass.

- Rev Comment 2 "6.3 detracts from the paper in that it is using speculation to justify an accumulation trend that may or may not be there. To investigate this, the results need to be evaluated together with the variability in the large scale atmospheric circulation, which is beyond the scope of

25 this already large paper."
    -> The section has been entirely removed.

- Rev Comment 3 " It is inappropriate at line 979 onwards to talk about rewriting the Antarctic ice sheet's negative sea-level contribution."

30     -> also removed.

- Rev Comment 1 : "The physical mechanisms that produce radar isochrones (internal reflection horizons, IRH's) are still unknown, yet thought to be linked to acidity and perhaps volcanic fallout horizons. In this paper, the dating of the ice cores has been well constrained by volcanic sulphate

35 horizons. However, the isochrones shown in Figure 2 and so crucial to the papers eventual conclusions are not located at depths where volcanic sulphate horizons occur. So the question remains as to whether they are also the smoothing product of postdepositional processes that concatenate IRH's at a similar depth interval."
    -> This is a problem inherent to the radar physics which is not specific to our paper. Interpretation

40 of radar reflectors has so far remained mostly heuristic and implicitly considered as such in all of the literature about radar data. We just give some justifications (last but one paragraph of Section 2.1) and I do not really see what else we could do in the frame of such a paper. The fact that the isochrones result from post-depositional smoothing process is probable. Considering the associated error as implicitly accounted in the error due to the representativeness of single ice cores (see Sec-

45 tion 4.3) does not seem irrealistic as we did and which led us to propose the corresponding 8 yr uncertainty. Again, we insist on the fact that this study is amongst the only one that proposes such a detailed error budget analysis on radar data. We feel that as rigorous as possible the way we have followed constitutes a significant progress in this so-far neglected issue.

50

Here is the list of the attached files :

55 Correction-1col-sub3.tex : Tex file for the 3rd version (one column)
Correction-1col-sub3.pdf : corresponding .pdf file
Correction-2col-sub3.pdf : the 2-column version pdf file
diff.pdf : pdf difference file showing the changes since the previous submission

60

Sincerely,

65
Emmanuel Le Meur

[revised manuscript text omitted]